# RoBlock: Wide and Deep Scaling of Recommenders via Embedding Collapse Mitigation

## Abstract

Scaling recommendation models has emerged as a promising direction for advancing recommender systems, yet they face a fundamental challenge: *embedding rank collapse*. This phenomenon, rooted in the intrinsic properties of feature interaction modules, causes embedding matrices to lose representational capacity as models scale, severely limiting their effectiveness. Existing solutions primarily focus on width-wise scaling via *multi-embedding*, which parallelizes multiple embedding tables and has shown success in alleviating collapse. However, these methods configure only the initial embedding layer and fail to address *depth-wise* embedding collapse, which intensifies with increasing model depth and restricts the benefits of deeper architectures. We propose **RoBlock**, a stackable building block that delivers dimensionally-robust embeddings by mitigating collapse across width and depth. RoBlock integrates three key components: (1) **spectrum rebalancing** through rank-1 update normalization to restore the spectrum distribution of embedding matrices, (2) an **embedding decoupler** guided by the Hilbert–Schmidt Independence Criterion (HSIC) to extract independent embedding components while preserving spectrum (dimensional) robustness, and (3) **embedding regeneration** via a field-wise multi-head router to regenerate non-collapsed embedding sets, achieving the benefits of multi-embedding within each block. Theoretical analysis establishes that RoBlock effectively mitigates embedding collapse, providing a principled foundation for scalable recommendation models. Extensive experiments across multiple datasets further demonstrate that RoBlock consistently alleviates embedding collapse across layers and delivers significant performance gains over the baselines, with improvements growing as model width and depth increase. The code is accessible at the anonymous link: https://anonymous.4open.science/r/RoBlock-2F8A.

## 1 Introduction

Recommender systems are a core application of machine learning, aiming to predict user–item interactions from massive multi-field categorical data (Zhang et al., 2016). They are now indispensable in daily life, powering applications in e-commerce, social media, news feeds, and music streaming. Researchers have advanced deep learning–based recommendation models that flexibly capture feature representations, leading to their successful deployment in a broad range of real-world applications.

Inspired by the success of large foundation models (Radford et al., 2021; Achiam et al., 2023; Rombach et al., 2022; Kirillov et al., 2023), scaling up model size has become a natural direction for recommender systems as well (Zhang et al., 2024a;b; Guo et al., 2024a; Wang et al., 2025a). Yet, counterintuitively, the embedding layers, which are arguably the most performance-critical component of recommendation models (Guo et al., 2017; Wang et al., 2021; Lian et al., 2018), are still configured with very small dimensions (e.g., 10 in open benchmarks (Zhu et al., 2022; 2021)). A key obstacle is the recently identified phenomenon of embedding rank collapse (Guo et al., 2024b; Pan et al., 2024; Chen et al., 2024; Zhang et al., 2025), which originates from feature interaction modules and forces embeddings to lie in low-rank subspaces. As a result, simply increasing embedding size yields diminishing or even negative returns.

To address this issue, the *multi-embedding* paradigm (Guo et al., 2024b) has emerged as a promising direction. By parallelizing multiple embedding layers, multi-embedding increases the effective

embedding size and alleviates embedding collapse, yielding strong empirical gains across diverse models (Lin et al., 2024; Liu et al., 2024; Wang et al., 2025b). However, multi-embedding primarily expands model *width*, alleviating collapse in the initial embedding layer. As a result, prior works typically adopt "shallow" architectures, where a multi-embedding layer is simply followed by a feature interaction layer and output projection. In contrast, modern recommendation models increasingly rely on stacking multiple layers, a paradigm we refer to as the *depth-wise* scaling regime. Our experiments on two representative depth-wise models, DHEN (Zhang et al., 2022) and Wukong (Zhang et al., 2024a), reveal that embedding collapse persists across layers, even when multi-embedding is applied (Figure 1), showing that embedding collapse in deep architectures remains underexplored.

In this work, we propose **RoBlock**, a stackable building block that mitigates embedding collapse across width and depth, enabling recommendation models to achieve dimensionally-robust scaling. Starting from the embedding matrix, RoBlock first applies rank-1 update normalization (Yu et al., 2020) to rebalance the embedding spectrum and enhance representational capacity. It then employs an embedding decoupler, regularized by the Hilbert–Schmidt independence criterion (HSIC) (Gretton et al., 2005; 2007), to extract independent components while preserving spectrum robustness. Next, a field-wise multi-head router regenerates expressive embedding sets, providing the benefits of multi-embedding at each block. We further employ heterogeneous functions for the feature interaction modules, following recent advances (Zeng et al., 2025; Zhang et al., 2022). Finally, the outputs are aggregated to form the input for the next RoBlock layer, yielding a unified, scalable framework.

Our theoretical analysis shows that RoBlock preserves embedding spectrum and effectively mitigates collapse at each layer. Complementary experiments across diverse datasets confirm that RoBlock consistently alleviates collapse and achieves significant performance gains over the baselines, with larger benefits for wider and deeper models. Together, these results position RoBlock as a principled paradigm for scalable recommendation systems. Our contributions are summarized as:

- *Methodological Innovation.* We propose RoBlock, a modular building block that mitigates embedding collapse across width and depth, establishing a new paradigm for scalable recommendation systems with dimensionally robust embeddings.

- *Theoretical Analysis.* We theoretically establish the spectrum (dimensional) robustness and rank-preserving properties of RoBlock, offering a solid view of its effectiveness.

- *Extensive Experiments.* We conduct extensive evaluations on benchmark datasets, demonstrating RoBlock's consistent performance gains and scalability benefits.

## 2 REVISITING EMBEDDING COLLAPSE: FROM WIDTH TO DEPTH

Recommendation models aim to predict a user action based on features drawn from multiple fields. Formally, consider $n$ fields, where the $i$-th field is denoted as $\mathcal{X}_i$, and define the joint feature space as $\mathcal{X} = \mathcal{X}_1 \times \mathcal{X}_2 \times \cdots \times \mathcal{X}_n$. Let $\mathcal{Y}$ denote the prediction space; the goal of a recommendation model is then to learn a mapping from $\mathcal{X}$ to $\mathcal{Y}$. Most mainstream recommendation models (Rendle, 2010; Guo et al., 2017; Zhang et al., 2022; Wang et al., 2021; Lian et al., 2018; Zhang et al., 2024a; He & Chua, 2017; Lin et al., 2024) rely on two key components: *(i) embedding layers*, which encode raw features $X \in \mathcal{X}$ into an embedding matrix $E \in \mathbb{R}^{n \times d}$, with $d$ denoting the embedding size, and *(ii) feature interaction modules*, which model cross-field interactions on $E$. The resulting interaction outputs can then be fed into prediction modules, such as MLPs, for downstream tasks.

**Embedding Collapse and Mitigation via Multi-Embedding.** Dimensional collapse refers to models degenerating into trivial solutions that map all inputs to the same constant (Hua et al., 2021), which can be quantified via the spectrum of learned representations (Jing et al., 2022). In recommendation models, a related problem termed *embedding collapse* has recently been identified: the embedding matrix becomes approximately low-rank with several near-zero singular values (Guo et al., 2024b; Pan et al., 2024; Chen et al., 2024; Zhang et al., 2025). To quantify this effect, Guo et al. (2024b) proposed the metric *information abundance* (IA), which serves as a robust indicator of collapse severity. Given the singular values $\{\sigma_j\}_{j=1}^{\min(n,d)}$ of the embedding $E \in \mathbb{R}^{n \times d}$, the IA of $E$ is defined as

$$\text{IA}(E) = \frac{\sum_{i=1}^{\min(n,d)} |\sigma_i|}{\max_i |\sigma_i|} \in [1, \min(n,d)],$$

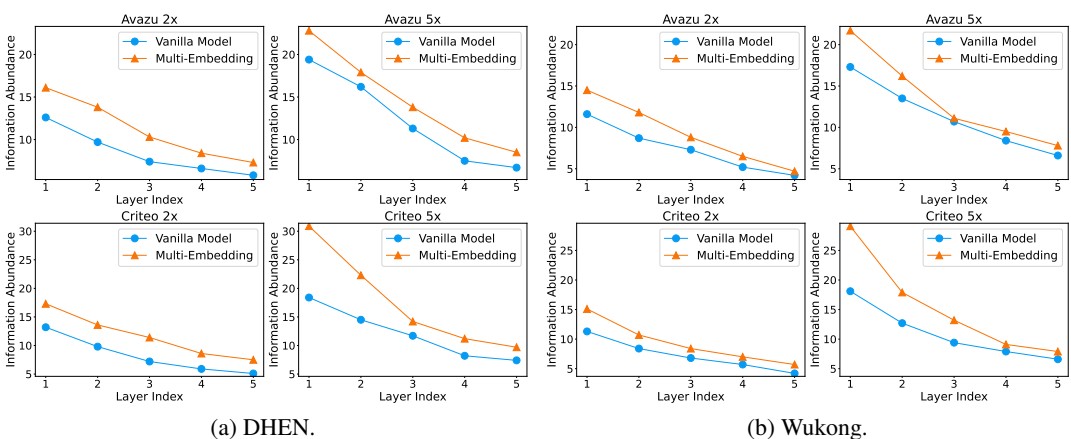

(a) DHEN.  (b) Wukong.

Figure 1: Embedding collapse, measured by information abundance, across layers for embedding sizes of $2\times$ and $5\times$ the base size. Each setting includes both vanilla and multi-embedding cases.

Among recent remedies for embedding collapse, the *multi-embedding* mechanism has gained notable traction (Guo et al., 2024b). By parallelizing multiple embedding layers $\{E_i\}_{i=1}^{m}$, multi-embedding effectively enlarges the embedding space. This enlargement alleviates embedding collapse with an amplified IA, yielding consistent performance gains across diverse recommendation tasks (Lin et al., 2024; Liu et al., 2024; Wang et al., 2025b; Su et al., 2024). However, vanilla multi-embedding is confined to the initial embedding stage, mapping raw features $X$ into embedding $E$ and expanding model capacity only in *width*. As a result, its use is largely restricted to "shallow" architectures, leaving its potential for deep recommendation models underexplored.

**Scalable Recommendation Models and Depth-wise Embedding Collapse.** Inspired by the success of large foundation models, scaling up model capacity has emerged as a central direction for modern recommender systems, with increasing focus on deeper architectures. Representative models such as DHEN (Zhang et al., 2022) and Wukong (Zhang et al., 2024a) demonstrate the promise of large-scale recommendation models.

However, those deep recommendation models can still suffer from embedding collapse, and vanilla multi-embedding provides only limited relief. Specifically, collapse occurs in two forms: *width-wise* and *depth-wise*. While multi-embedding effectively mitigates width-wise collapse, it is largely ineffective against depth-wise collapse. We empirically validate this phenomenon on DHEN and Wukong using the Criteo (Jean-Baptiste Tien, 2014) and Avazu (Steve Wang, 2014) datasets, comparing vanilla and multi-embedding variants. As shown in Figure 1, while multi-embedding consistently increases IA across layers compared to the vanilla models, IA in deeper layers still declines sharply. This indicates that embeddings in deeper layers continue to collapse similarly to the vanilla models, suggesting that multi-embedding offers only a limited remedy for depth-wise collapse. This depth-wise collapse poses a critical challenge for deep recommendation models, motivating the development of our RoBlock that mitigates both width-wise and depth-wise collapse.

## 3 ROBLOCK

In this section, we introduce our **RoBlock**, which mitigates embedding collapse across both width and depth, thereby ensuring dimensional robustness. Figure 2 provides an overview of the RoBlock architecture, followed by a detailed examination of each module in the subsequent subsections.

**Initial embedding layer.** Let $X^{\text{raw}}$ be the multi-field raw features, $n$ the number of fields, and $d$ the embedding dimension. Prior to the RoBlock, following vanilla multi-embedding (Guo et al., 2024b), we first apply $m$ embedding layers to transform $X^{\text{raw}}$ into $\{E_i^{(0)}\}_{i=1}^{m}$, each of shape $n \times k$. Here, $k$ is the *base size*, corresponding to the common dimension in recommendation models. For simplicity, we use $m = \lfloor d/k \rfloor + 1$ to ensure sufficient capacity, but both $m$ and $k$ can be chosen

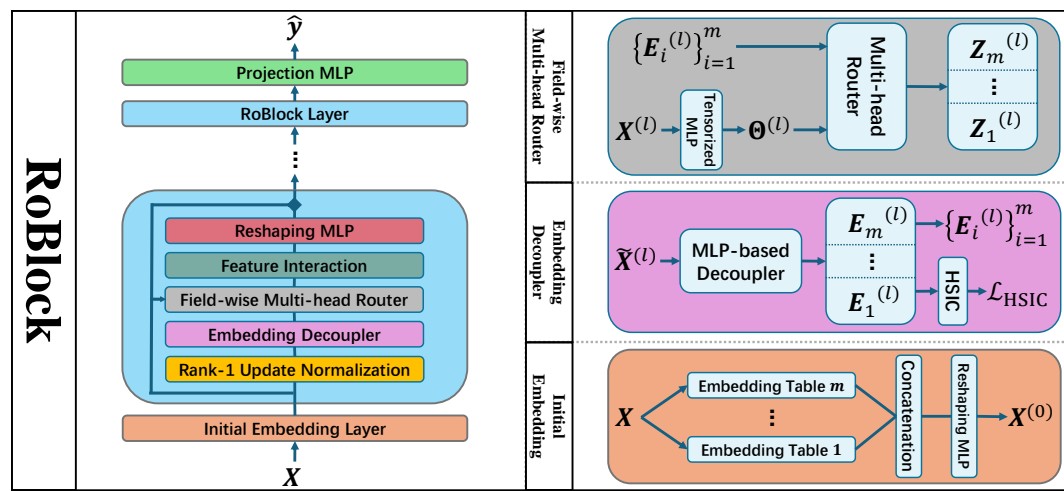

Figure 2: Overview of the RoBlock, with the internal structure of a single layer highlighted in blue.

flexibly. The embeddings are concatenated and transformed via a reshaping MLP, $f^{(0)}$ to produce

$$X^{(0)} = f^{(0)}([E_1^{(0)}, E_2^{(0)}, \dots, E_m^{(0)}]) \in \mathbb{R}^{n \times d}, \tag{1}$$

which serves as input to the first RoBlock. In the following, without loss of generality, we focus on the $l$-th RoBlock, which takes $X^{(l-1)} \in \mathbb{R}^{n \times d}$ as input and produces $X^{(l)} \in \mathbb{R}^{n \times d}$ as output.

## 3.1 ENHANCING EMBEDDING VIA RANK-1 UPDATE NORMALIZATION

Received the embedding matrix $X^{(l-1)}$, RoBlock first applies **rank-1 update normalization** (Yu et al., 2020), an algorithm motivated by the power method (Stoer et al., 1980). Initialized with a random vector $v^{(0)} \sim \mathcal{N}(0, I)$, where $I$ is $d$-dimensional identity matrix, rank-1 update normalization with $t$ iterations refines $X^{(l-1)}$ as:

$$\tilde{X}^{(l-1)} = X^{(l-1)} - \frac{X^{(l-1)} v^{(t)} v^{(t)\top}}{\|v^{(t)}\|_2^2}; \quad v^{(j)} = (X^{(l-1)})^\top X^{(l-1)} v^{(j-1)} \quad \text{for} \quad j = 1, \dots, t. \tag{2}$$

This operation produces an embedding $\tilde{X}^{(l-1)}$ with a re-balanced spectrum, a property validated by many prior studies like Chen et al. (2022); Wang et al. (2024). Thus, $\tilde{X}^{(l-1)}$ exhibits increased information abundance compared to $X^{(l-1)}$ (see Section 4.1 for theoretical analysis), providing a higher-quality input for subsequent processing. We treat this step as a training augmentation that regularizes the model and improves overall efficacy, with complexity analysis in Appendix A.4.4.

## 3.2 EXTRACTING INDEPENDENT COMPONENTS VIA EMBEDDING DECOUPLER

Given the enhanced embedding $\tilde{X}^{(l-1)}$, RoBlock leverages an **embedding decoupler** composed of $m$ parallel MLPs to decompose $\tilde{X}^{(l-1)}$ into a set of independent, low-dimensional embeddings $\{E_i^{(l-1)}\}_{i=1}^m$. Each $E_i^{(l-1)}$ has shape $n \times k$, matching the dimensionality of the embedding tables in the initial embedding layer. Mutual independence among $\{E_i^{(l-1)}\}_{i=1}^m$ is enforced using the Hilbert–Schmidt independence criterion (HSIC) (Gretton et al., 2005; 2007) as an auxiliary loss:

$$\mathcal{L}_{\text{HSIC}}^{(l-1)} = \frac{2}{m^2 - m} \sum_{\substack{p < q \\ p,q \in [1,m]}} \frac{1}{(n-1)^2} \operatorname{tr}\left( K(E_p^{(l-1)}) H L(E_q^{(l-1)}) H \right). \tag{3}$$

Here, $H = I - \frac{1}{n} \mathbf{1}\mathbf{1}^\top$ denotes the centering matrix, and $K(\cdot)$ and $L(\cdot)$ are kernel functions. We adopt a linear kernel, so both $K$ and $L$ reduce to simple inner products. HSIC has proven effective for measuring feature independence and has been widely used as an optimization objective in various machine learning applications (Greenfeld & Shalit, 2020; Kumagai et al., 2022; Chen et al., 2025).

Furthermore, the resulting $\{E_i^{(l-1)}\}_{i=1}^m$ exhibit *spectrum (dimensional) robustness* when concatenated (see Section 4.2), showing that the decoupler extracts independent components in a rank-preserving manner, ensuring the concatenated embeddings retain high informational capacity.

### 3.3 Embedding Regeneration for Non-Collapsed Feature Interaction

To provide expressive inputs for feature interaction, RoBlock employs a **field-wise multi-head router** that regenerates embeddings in a MoE-like fashion. Concretely, a tensorized MLP $g^{(l-1)}$ maps the input $X^{(l-1)}$ to a weight tensor $\Theta^{(l-1)} \in \mathbb{R}^{n \times m \times m}$, which determines how the independent components $\{E_i^{(l-1)}\}_{i=1}^m$ are combined. The regenerated embeddings $\{Z_i^{(l-1)}\}_{i=1}^m$, serving as inputs to the interaction modules, are computed as

$$Z_i^{(l-1)} = \sum_{j=1}^n \mathrm{diag}(\Theta_{:ji}^{(l-1)})E_j^{(l-1)} ; \quad \Theta^{(l-1)} = g^{(l-1)}(X^{(l-1)}). \tag{4}$$

Unlike (Yin et al., 2025), which adopts a generative paradigm for feature interaction in shallow models, RoBlock focuses on regenerating embeddings to enable adaptive learning of expressive inputs for interaction modules in deep architectures. Section 4.3 further confirms that this regeneration preserves non-collapsed embeddings across depths and effectively mitigates interaction collapse.

Moreover, following recent advances in **heterogeneous feature interaction** (Zeng et al., 2025; Zhang et al., 2022; Wang et al., 2025b), which combine classical interaction functions into ensembles for better downstream performance, RoBlock processes the regenerated embeddings through multiple preselected interaction functions $\{I_i^{(l-1)}(\cdot)\}_{i=1}^m$. The outputs are then concatenated and reshaped via an MLP $f^{(l-1)}$ to restore a unified shape $n \times d$ for the next RoBlock input $X^{(l)}$:

$$X^{(l)} = X^{(l-1)} + f^{(l-1)}([I_1^{(l-1)}(Z_1^{(l-1)}), I_2^{(l-1)}(Z_2^{(l-1)}), \ldots, I_m^{(l-1)}(Z_m^{(l-1)})]). \tag{5}$$

### 3.4 Optimization

Let $L$ denote the number of RoBlock layers. We aggregate outputs from all layers via simple averaging and pass the result through a projection MLP $f^*$ to produce the final prediction $y$, capturing multi-level feature interactions while stabilizing training. Following standard practice in recommendation benchmarks (Zhu et al., 2022; 2021), we adopt binary cross-entropy as the base loss and further incorporate HSIC regularization with trade-off $\beta$, yielding the overall training objective:

$$\mathcal{L} = \mathcal{L}_{\mathrm{BCE}}(y_{\mathrm{true}}, y) + \beta \sum_{l=1}^L \mathcal{L}_{\mathrm{HSIC}}^{(l-1)}; \quad y = f^*\left(\frac{1}{L}\sum_{l=1}^L X^{(l)}\right). \tag{6}$$

## 4 Theoretical Analysis

We analyze the embedding spectrum at every distinct module of the RoBlock. Our focus lies on three central aspects: *(i) **How rank-1 update normalization re-balance embedding spectrum**; (ii) **How the embedding decoupler preserves spectrum (dimensional) robustness**; (iii) **How field-wise gating prevents collapse in feature interaction**.* We follow the same notation as in earlier sections but omit layer indices, as the analysis applies generally and is not restricted to any specific case.

### 4.1 Re-balancing Spectrum via Rank-1 Update Normalization

RoBlock applies rank-1 update normalization to the input $X$, yielding an enhanced embedding $\tilde{X}$ with a rebalanced spectrum. This effect is formally captured by the following proposition:

**Proposition 4.1.** *Let $\tilde{X}$ denote the enhanced embedding obtained via Equation 2 at the $t$-th iteration, starting from a random vector $v^{(0)} \sim \mathcal{N}(0, I)$. Then $\mathbb{E}_{v^{(0)} \sim \mathcal{N}(0,I)}(\tilde{X}) = U\tilde{\Sigma}V^\top$ exhibits a rebalanced spectrum, where $\sigma_1 \geq \sigma_2 \geq \cdots \geq \sigma_d$ denote the ordered singular values of $X$,*
$\tilde{\Sigma} = \mathrm{diag}\left[(1-\lambda_1(t))\sigma_1, (1-\lambda_2(t))\sigma_2, \cdots, (1-\lambda_d(t))\sigma_d\right], \lambda_i(t) = \mathbb{E}_{y \sim \mathcal{N}(0;I)}\left(\frac{(y_i\sigma_i^{2t})^2}{\sum_{l=1}^d (y_l\sigma_l^{2t})^2}\right),$
*and $y = V^\top v^{(0)}$. Since $0 \leq 1-\lambda_1(t) \leq 1-\lambda_2(t) \leq \cdots \leq 1-\lambda_d(t) \leq 1$ for $\sigma_1 \geq \sigma_2 \geq \cdots \geq \sigma_d$, so the $(1-\lambda_i)$ gets smaller or larger as $\sigma_i$ gets larger or smaller, respectively.*

The proof is provided in Appendix A.1. Proposition 4.1 shows that applying rank-1 update normalization to $X$ reduces large singular values while amplifying smaller ones in the resulting $\tilde{X}$. Consequently, the spectrum of $\tilde{X}$ becomes more balanced. This rebalancing also leads to an increase in information abundance.

To illustrate this effect, Figure 3 presents a numerical example using a low-rank random matrix with $n = 10{,}000$ and $d = 10$, where rank-1 updates are applied for one and two iterations. The plots show the distribution of singular values (**normalized by the maximum**) together with the corresponding information abundance.

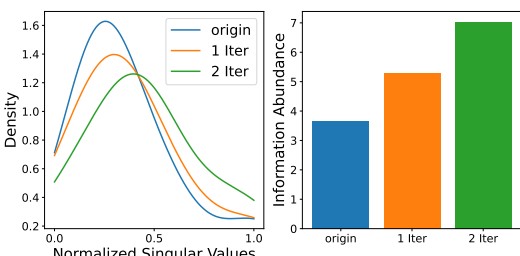

Figure 3: Effect of rank-1 update normalization on the singular value distribution and the corresponding information abundance.

With each iteration, the density of large singular values ($\lambda \in [0.5, 1]$) increases, while that of smaller ones ($\lambda \in [0, 0.5]$) decreases. This density shift indicates a more balanced spectrum—by contrast, the original matrix exhibits a high density of small singular values, reflecting rank collapse—further confirming that rank-1 update normalization effectively enhances the embedding quality.

## 4.2 Spectrum (Dimensional) Robustness of Embedding Decoupler

RoBlock employs the embedding decoupler to generate a set of $m$ embeddings, regularized via the HSIC. These embeddings, beyond the mutual independence enforced by HSIC (Gretton et al., 2005; 2007), also exhibit crucial *spectrum (dimensional) robustness*, as formally stated below:

**Proposition 4.2.** *Let $E_i = E_i(\tilde{X})$, for $i = 1, \ldots, m$, with $mk \geq d$, denote the embeddings generated by the decoupler and regularized using HSIC, and let the concatenated embedding be $E(\tilde{X}) = [E_1(\tilde{X}); E_2(\tilde{X}); \ldots; E_m(\tilde{X})]$. Under exact HSIC regularization, we have $\mathrm{rank}(E(\tilde{X})) = \sum_{i=1}^{m} \mathrm{rank}(E_i(\tilde{X}))$. Moreover, under perturbation or approximate HSIC regularization, let the block covariance be $\Sigma_{\mathrm{blk}} = \mathbb{E}[\mathrm{vec}(E(\tilde{X}))\,\mathrm{vec}(E(\tilde{X}))^\top]$, and let $\Sigma = \Sigma_{\mathrm{blk}} + \Delta$, where $\Delta$ denotes the perturbation. If $\Sigma_{\mathrm{blk}}$ has $r$ eigenvalues bounded below by $\gamma > 0$ and $\|\Delta\|_2 < \gamma$, then $\Sigma$ retains at least $r$ eigenvalues greater than $\gamma - \|\Delta\|_2 > 0$. Hence, the numerical rank of $\Sigma$ is preserved up to the spectral norm of the perturbation $\Delta$, implying that $\Sigma_{\mathrm{blk}}$ and, consequently, $E(\tilde{X})$ maintains robustness of its spectrum.*

The proof is provided in Appendix A.2. Proposition 4.2 establishes that HSIC regularization prevents the embedding sets produced by the decoupler from collapsing into redundant subspaces. By enforcing independence, the effective rank of the concatenated embedding equals the sum of the ranks of its components. Thus, each embedding contributes complementary information, and the overall representational capacity scales additively with the number of components. Moreover, even when HSIC is only approximately minimized, the perturbation bound guarantees spectrum (dimensional) robustness. Consequently, the embedding decoupler provably extracts non-redundant embedding sets while preserving representation capacity.

## 4.3 Mitigating Interaction Collapse through Field-wise Multi-head Router

Previous research has confirmed that feature interaction modules play a notorious role in causing embedding collapse, termed interaction collapse (Guo et al., 2024b). We show that this collapse can be effectively mitigated by applying our field-wise multi-head router to regenerate the embeddings fed into the feature interaction modules, as formalized below:

**Proposition 4.3.** *Let $\theta_{j,i} > 0$ be the per-row weights generated by an MLP-parameterized field-wise gating mechanism. Then the gated embedding $Z = \sum_{i=1}^{m} D_i E_i$, with $D_i = \mathrm{diag}(\theta_{1,i}, \ldots, \theta_{n,i}) \in \mathbb{R}^{n \times n}$, which serves as the input to a feature interaction module, satisfies*

$$\max_i \mathrm{rank}(E_i) \leq \mathrm{rank}(Z) \leq \min\left(k, \sum_{i=1}^{m} \mathrm{rank}(E_i)\right). \tag{7}$$

*Consequently, $Z$ is guaranteed to be high-rank, ensuring that downstream feature interaction modules operate on a non-collapsed embedding space.*

Table 1: Test AUC of shallow models on Criteo (higher is better). Bold values indicate the best performance among vanilla single embedding (SE) and multi-embedding (ME) settings. **Complete results with standard deviation (std) are provided in Appendix** A.4.3.

| Model | | Criteo | | | | |
|---|---|---|---|---|---|---|
| | | base | 2× | 3× | 5× | 10× |
| DNN | SE | 0.810924 | 0.810734 | 0.810921 | 0.810615 | 0.810486 |
| | ME | | 0.811251 | 0.811379 | 0.811498 | 0.811684 |
| Crossnet | SE | 0.811615 | 0.811632 | 0.811401 | 0.811522 | 0.811243 |
| | ME | | 0.811903 | 0.812084 | 0.812221 | 0.812453 |
| NFM | SE | 0.808619 | 0.808521 | 0.808622 | 0.808473 | 0.808235 |
| | ME | | 0.808843 | 0.809041 | 0.809153 | 0.809332 |
| CIN | SE | 0.811711 | 0.811681 | 0.811714 | 0.811702 | 0.811459 |
| | ME | | 0.811991 | 0.812175 | 0.812259 | 0.812483 |
| xDeepFM | SE | 0.813482 | 0.813401 | 0.813221 | 0.813333 | 0.812977 |
| | ME | | 0.813662 | 0.813751 | 0.813922 | 0.814209 |
| DCNv2 | SE | 0.813148 | 0.813042 | 0.813151 | 0.812998 | 0.812757 |
| | ME | | 0.813364 | 0.813477 | 0.813622 | 0.813952 |
| **RoBlock(Ours)** | | **0.813824** | **0.814011** | **0.814134** | **0.814242** | **0.814639** |

Proof is provided in Appendix A.3. Proposition 4.3 shows that the field-wise gating (router) mechanism yields a gated embedding $Z$ whose rank is guaranteed to be at least the maximum rank of the independent components. This property is critical for mitigating interaction collapse: as shown by Guo et al. (2024b), collapse in interaction modules originates from collapse in their inputs. In contrast, our gated embedding $Z$ maintains a high rank, ensuring that subsequent interaction modules operate on non-collapsed embedding. The corollaries in Appendix A.3 further confirm that this rank preservation extends to the outputs of feature interactions across diverse models, conveying informative and high-quality inputs for the next RoBlock.

## 5 EXPERIMENTS

In this section, we aim to address these research questions: **RQ1:**Does RoBlock consistently outperform existing baselines on public datasets? **RQ2:**How effectively does RoBlock mitigate embedding collapse? **RQ3:**How does scaling model width and depth affect RoBlock's performance?

### 5.1 SETUP

We evaluate on two large-scale recommendation benchmarks, Criteo (Jean-Baptiste Tien, 2014) and Avazu (Steve Wang, 2014), both sourced from the FuxiCTR (Zhu et al., 2021) library. Model performance is measured by AUC. Baselines are grouped into two categories: *(i) shallow architectures*, which follow the standard "embedding–interaction–projection" paradigm, and *(ii) deep architectures*, which stack multiple layers of the shallow design. For the shallow group, we include DNN, CrossNet (Wang et al., 2021), NFM (He & Chua, 2017), CIN (Lian et al., 2018), xDeepFM (Lian et al., 2018), and DCNv2 (Wang et al., 2021), each evaluated in both vanilla and multi-embedding variants. For the deep group, we include DHEN (Zhang et al., 2022) and Wukong (Zhang et al., 2024a), evaluated only with their multi-embedding variants. All baselines are trained with embedding dimensions scaled by 2×, 3×, 5×, and 10× relative to the base size. We use the 8/1/1 train/validation/test split, consistent with FuxiCTR. More details are shown in Appendix A.4.

### 5.2 RQ1: EVALUATING ROBLOCK AGAINST BASELINES

**Comparison with Shallow Architectures.** For a fair comparison with shallow architectures, we set RoBlock's depth to 1, thereby reducing it to the standard "embedding–interaction–projection" paradigm used by the shallow baselines. As shown in Tables 1 and 2, RoBlock consistently outperforms all baselines, with its advantage widening as the embedding size increases. Notably, even in settings where shallow models already benefit from multi-embedding (Guo et al., 2024b), RoBlock delivers substantially larger gains, demonstrating a more effective utilization of multi-embedding.

Table 2: Test AUC of shallow models on Avazu. **Complete results with standard deviation (std) are provided in Appendix** A.4.3.

| Model | | Avazu | | | | |
|---|---|---|---|---|---|---|
| | | base | 2× | 3× | 5× | 10× |
| DNN | SE | 0.789217 | 0.789201 | 0.789041 | 0.789144 | 0.788962 |
| | ME | | 0.789577 | 0.789663 | 0.789822 | 0.790041 |
| Crossnet | SE | 0.790102 | 0.790043 | 0.789954 | 0.790022 | 0.789828 |
| | ME | | 0.790382 | 0.790447 | 0.790613 | 0.790861 |
| NFM | SE | 0.787988 | 0.787964 | 0.788021 | 0.787898 | 0.787641 |
| | ME | | 0.788277 | 0.788464 | 0.788563 | 0.788912 |
| CIN | SE | 0.792111 | 0.792073 | 0.792181 | 0.792123 | 0.791921 |
| | ME | | 0.792374 | 0.792511 | 0.792649 | 0.792902 |
| xDeepFM | SE | 0.793427 | 0.793356 | 0.793421 | 0.793303 | 0.793021 |
| | ME | | 0.793688 | 0.793825 | 0.793961 | 0.794201 |
| DCNv2 | SE | 0.794033 | 0.794041 | 0.794108 | 0.794133 | 0.794051 |
| | ME | | 0.794173 | 0.794272 | 0.794361 | 0.794522 |
| **RoBlock(Ours)** | | **0.794929** | **0.795432** | **0.796114** | **0.796502** | **0.796825** |

Table 3: Test AUC of deep models across varying depths. **Complete results, including standard deviations (std) as well as additional depths and sizes, are provided in the Appendix** A.4.3.

| Model | #layers | Criteo | | | Avazu | | |
|---|---|---|---|---|---|---|---|
| | | 2× | 5× | 10× | 2× | 5× | 10× |
| DHEN | 2 | 0.813584 | 0.813873 | 0.814207 | 0.794096 | 0.794396 | 0.794611 |
| | 5 | 0.813789 | 0.814057 | 0.814349 | 0.794278 | 0.794556 | 0.794789 |
| Wukong | 2 | 0.813427 | 0.813631 | 0.813680 | 0.793822 | 0.793980 | 0.794100 |
| | 5 | 0.813632 | 0.813804 | 0.813859 | 0.793985 | 0.794184 | 0.794293 |
| **RoBlock(Ours)** | 2 | **0.814065** | **0.814279** | **0.814689** | **0.795474** | **0.796542** | **0.796865** |
| | 5 | **0.814258** | **0.814480** | **0.814802** | **0.795657** | **0.796725** | **0.797022** |

This observation aligns with our theoretical analysis (Section 4): RoBlock's spectrum (dimensional) robustness and collapse-mitigation mechanisms preserve the benefits of multi-embedding in the initial layer, resulting in more pronounced performance improvements.

**Comparison with Deep Architectures.** As shown in Table 3, RoBlock achieves higher performance gains than DHEN and Wukong, and these gains amplify as both embedding size and network depth increase. This indicates that RoBlock harnesses the benefits of both width and depth more effectively, highlighting its strength as an advanced paradigm for scalable recommendation models. In the next subsection, we analyze the scaling behavior of RoBlock w.r.t both depth and width.

## 5.3 RQ2: MITIGATING LAYER-WISE EMBEDDING COLLAPSE

To evaluate RoBlock's ability to mitigate embedding collapse and its impact on model performance, guided by the theoretical results in Section 4, we conduct ablation experiments on its key anti-collapse components: rank-1 update normalization (RU) and HSIC regularization. We report both test AUC and layerwise information abundance, with DHEN and Wukong included as additional baselines. Results are presented in Figures 4 and 5.

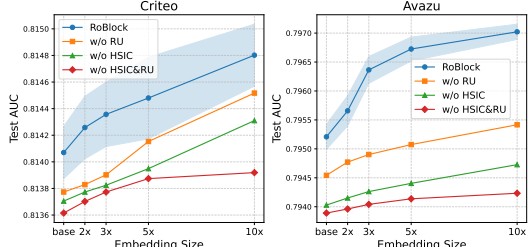

Figure 5: Comparison of test AUC performance across different RoBlock variants.

The results demonstrate that integrating these carefully designed modules substantially enhances RoBlock's information abundance. Without these modules ("w/o HSIC&RU"), RoBlock's IA slightly lags behind the baselines. Adding

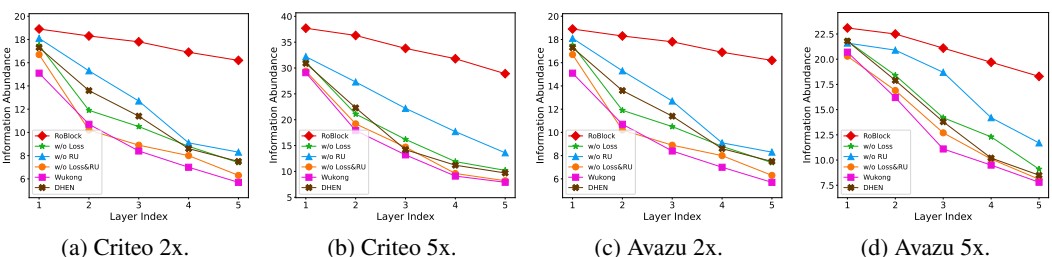

(a) Criteo 2x.  (b) Criteo 5x.  (c) Avazu 2x.  (d) Avazu 5x.

Figure 4: Comparison of layer-wise IA between RoBlock variants and depth-wise baselines.

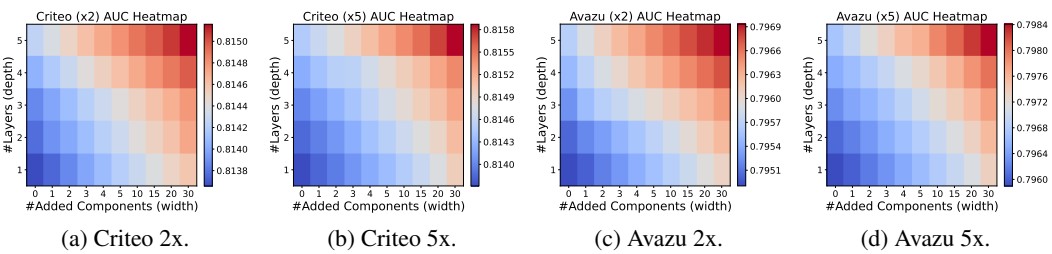

(a) Criteo 2x.  (b) Criteo 5x.  (c) Avazu 2x.  (d) Avazu 5x.

Figure 6: RoBlock test AUC across number of layers and added components. **Complete results with additional scaled sizes are presented in the Appendix** A.4.3.

each module incrementally improves IA, and the full design (denoted "RoBlock") achieves a significant gain, clearly surpassing all baselines. Notably, RoBlock maintains consistently high IA across all layers, while other models exhibit embedding collapse as early as the second, third, or fourth layer, highlighting the effectiveness of our anti-collapse design. Moreover, this increase in IA results in improved model performance, as shown in Figure 5, empirically validating the strong connection between mitigating embedding collapse and enhancing predictive performance.

### 5.4 RQ3: ANALYZING SCALING EFFECTS IN DEPTH AND WIDTH

To assess the scalability of RoBlock, we perform ablation studies examining its behavior under different model scales. Specifically, we visualize heatmaps of Test AUC across varying numbers of layers (reflecting model depth) and additional components beyond the default $m = \lfloor d/k \rfloor + 1$ configuration (reflecting model width). As shown in Figure 6, RoBlock consistently improves Test AUC as both depth and width increase, across multiple embedding sizes. These results demonstrate that RoBlock effectively leverages additional model capacity, validating its scalability as a general paradigm. Moreover, the performance improvements from scaling only depth or only width are substantially smaller than those from scaling both simultaneously. The results indicate that true scalability in recommendation models arises not from depth or width alone, but from their balanced combination. RoBlock's ability to mitigate collapse in both aspects translates this principle into practice, yielding synergistic gains and providing a solid basis for scalable recommendation systems.

### 5.5 ASSESSING ROBLOCK'S COMPUTATIONAL EFFICIENCY

We further evaluate the computational efficiency of RoBlock relative to baseline models, focusing on **GPU memory allocation** and **training time per epoch**. To provide an intuitive comparison under a challenging setting, all models are evaluated using the largest 10x scale-up configuration (embedding dimension $d$ be 100). For fairness, RoBlock is assessed in a 1-layer configuration when compared with shallow baselines, and across multiple layers when compared with deep architectures, consistent with the experimental setups in prior sections. All baselines use their multi-embedding variants to remain fully aligned with the preceding experimental setup. We present the main results here, with additional findings deferred to Appendix A.5.1.

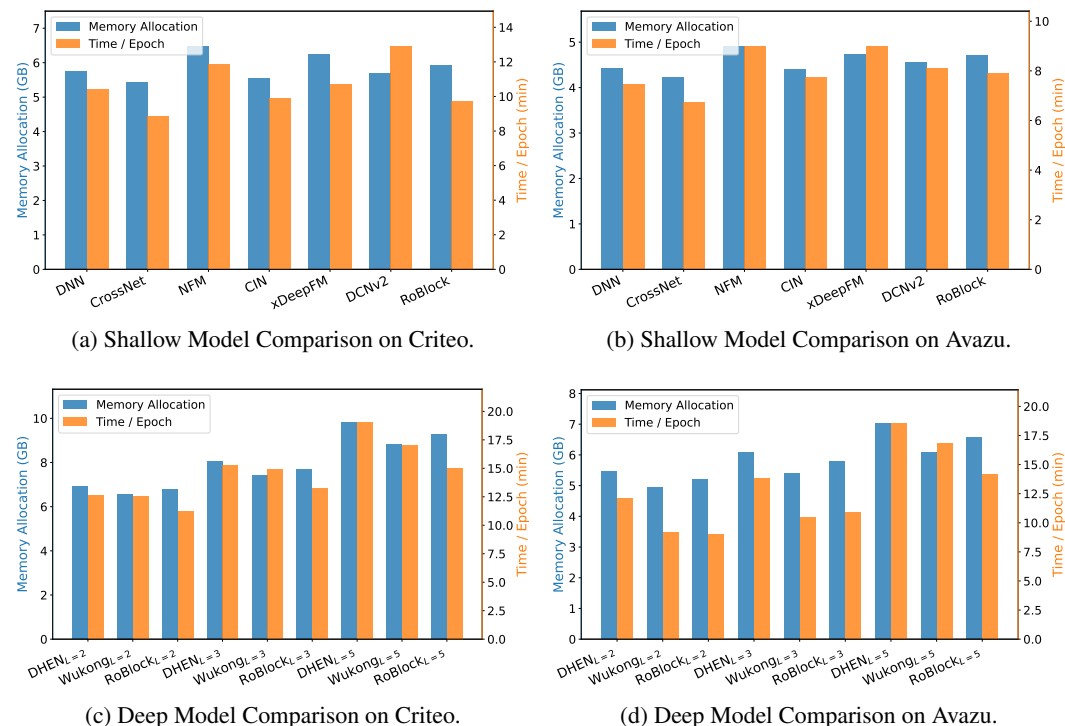

(a) Shallow Model Comparison on Criteo.

(b) Shallow Model Comparison on Avazu.

(c) Deep Model Comparison on Criteo.

(d) Deep Model Comparison on Avazu.

Figure 7: GPU memory allocation and training time of baselines and RoBlock. All models are evaluated under a 10× scale-up.

Figure 7 summarizes the results. The **blue** bars, which denote GPU memory usage, show that RoBlock consistently achieves leading memory efficiency, remaining competitive with both shallow and deep baselines. Similarly, the **orange** bars, representing training time per epoch, illustrate that RoBlock maintains strong training-time efficiency, matching or exceeding most baselines. These advantages can be attributed to three main factors: *(i)* RoBlock's modules are primarily configured as a lightweight MLP backbone; *(ii)* the low computational cost of the HSIC term and the rank-1 update normalization, as detailed in the complexity analyses in Appendix A.4.4 and Appendix A.5.2; and *(iii)* the simple feature interaction design, which reuses efficient components from prior work (e.g., adopting only the CrossNetv2 module from DCNv2).

Overall, these findings highlight RoBlock's practicality: it delivers strong predictive improvements while maintaining competitive memory usage, fast training, and efficient inference, showcasing strong potential for real-world deployment.

## 6 CONCLUSION

We study the embedding collapse problem in recommendation models, including the depth-wise collapse observed in modern deep architectures. To address collapse across both width and depth, we propose RoBlock, a stackable and scalable block that integrates rank-1 update normalization for embedding enhancement, a HSIC-regularized decoupler for extracting independent components, and a field-wise multi-head router to regenerate embeddings for heterogeneous feature interactions, resulting in a unified and scalable framework. Our theoretical analysis shows that each mechanism mitigates collapse and improves spectrum (dimensional) robustness, while experiments demonstrate substantial information abundance and superior predictive performance compared to baselines. These benefits are particularly pronounced in wider and deeper models, establishing RoBlock as a promising paradigm for scalable recommendation systems.

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

# A APPENDIX

## USE OF LLMs

This paper utilizes large language models (LLMs), including ChatGPT https://chat.openai.com/ and Gemini https://gemini.google.com/app, as general-purpose tools for writing assistance. This use of LLMs complies with the ICLR 2026 Author Guide https://iclr.cc/Conferences/2026/AuthorGuide.

## A.1 PROOF OF PROPOSITION 4.1

**Proposition 4.1.** *Let $\tilde{X}$ denote the enhanced embedding obtained via Equation 2 at the $t$-th iteration, starting from a random vector $v^{(0)} \sim \mathcal{N}(0, I)$. Then $\mathbb{E}_{v^{(0)} \sim \mathcal{N}(0,I)}(\tilde{X}) = U \tilde{\Sigma} V^{\top}$ exhibits*

*a rebalanced spectrum, where $\sigma_1 \geq \sigma_2 \geq \cdots \geq \sigma_d$ denote the ordered singular values of $X$,*
$\tilde{\Sigma} = \text{diag}\left[(1-\lambda_1(t))\sigma_1, (1-\lambda_2(t))\sigma_2, \cdots, (1-\lambda_d(t))\sigma_d\right]$, $\lambda_i(t) = \mathbb{E}_{y \sim \mathcal{N}(0;I)}\left(\frac{(y_i\sigma_i^{2t})^2}{\sum_{l=1}^{d}(y_l\sigma_l^{2t})^2}\right)$,
*and $y = V^\top v^{(0)}$. Since $0 \leq 1 - \lambda_1(t) \leq 1 - \lambda_2(t) \leq \cdots \leq 1 - \lambda_d(t) \leq 1$ for $\sigma_1 \geq \sigma_2 \geq \cdots \geq \sigma_d$,*
*so the $(1 - \lambda_i)$ gets smaller or larger as $\sigma_i$ gets larger or smaller, respectively.*

*Proof.* Let $X = U\Sigma V^\top$ be the SVD of the embedding, where $U$ and $V$ are the right and left orthonormal matrices that contain the singular vectors. As $v^{(t)} = (X^\top X)^t v^{(0)}$, we have:

$$v^{(t)} = (X^\top X)^t v^{(0)} = (V\Sigma^{2t}V^\top)v^{(0)}. \tag{8}$$

Plugging Equation (8) into Equation 2, we obtain:

$$
\begin{aligned}
\frac{Xv^{(t)}v^{(t)\top}}{\|v^{(t)}\|_2^2} &= \frac{U\Sigma V^\top(V\Sigma^{2t}V^\top)v^{(0)}v^{(0)\top}(V\Sigma^{2t}V^\top)}{v^0(V\Sigma^{2t}V^\top)(V\Sigma^{2t}V^\top)v^0} \\
&= \frac{U\Sigma^{2t+1}V^\top v^{(0)}v^{(0)\top}V\Sigma^{2t}V^\top}{v^{(0)\top}V\Sigma^{4t}V^\top v^{(0)}}.
\end{aligned} \tag{9}
$$

Let $y = V^\top v^{(0)}$ and $t \geq 1$ be the number of iterations, then we simplify Eq. equation 9 as:

$$\frac{Xv^{(t)}v^{(t)\top}}{\|v^{(t)}\|_2^2} = UQ(y,t)V^\top \tag{10}$$

$$\text{where } Q(y,t) = \frac{\Sigma^{2t+1}yy^\top\Sigma^{2t}}{y^\top\Sigma^{4t}y}. \tag{11}$$

Because $v^{(0)} \sim \mathcal{N}(0,I)$ and $V$ is a unitary matrix, we have $y = [y_1, \cdots, y_d] \sim \mathcal{N}(0,I)$. Moreover, note that the expectation of $Q$ is a diagonal matrix, we have $\mathbb{E}_{y \sim \mathcal{N}(0,I)}(Q(y,t)) = \text{diag}\left(q_1(t), q_2(t), \cdots, q(t)_d\right)$ with:

$$q_i(t) = \sigma_i\lambda_i(t) \text{ where } \lambda_i(t) = \mathbb{E}_{y \sim \mathcal{N}(0;I)}\left(\lambda_i'(y,t)\right),$$
$$\text{and } \lambda_i'(y,t) = \frac{(y_i\sigma_i^{2t})^2}{\sum_{l=1}^{d}(y_l\sigma_l^{2t})^2}. \tag{12}$$

For brevity, let us drop parameter $t$ where possible. We need to show that $1 \geq \lambda_i \geq \lambda_j \geq 0$ when $\sigma_i \geq \sigma_j$. Obviously, $1 \geq \lambda_i \geq 0$, thus we need to show that $\lambda_i - \lambda_j \geq 0$:

$$\lambda_i - \lambda_j = \mathbb{E}\left(\frac{(y_i\sigma_i^{2t})^2}{\sum_{l=1}^{d}(y_l\sigma_l^{2t})^2}\right) - \mathbb{E}\left(\frac{(y_j\sigma_j^{2t})^2}{\sum_{l=1}^{d}(y_l\sigma_l^{2t})^2}\right) \geq \sigma_j^{4t}\mathbb{E}\left(\frac{y_i^2 - y_j^2}{\sum_{l=1}^{d}(y_l\sigma_l^{2t})^2}\right) = 0. \tag{13}$$

Because

$$\mathbb{E}_{y \sim \mathcal{N}(0;I)}\left(\frac{y_i^2 - y_j^2}{\sum_{l=1}^{d}(y_l\sigma_l^{2t})^2}\right) = \mathbb{E}_{y \sim \mathcal{N}(0;I)}\left(\frac{y_i^2}{\sum_{l=1}^{d}(y_l\sigma_l^{2t})^2}\right) - \mathbb{E}_{y \sim \mathcal{N}(0;I)}\left(\frac{y_j^2}{\sum_{l=1}^{d}(y_l\sigma_l^{2t})^2}\right) = 0 \tag{14}$$

due to $y_i$, $y_j$ being i.i.d. random variables sampled from the normal distribution, inequality equation 13 holds, thereby concluding the proof. $\square$

### A.2 PROOF OF PROPOSITION 4.2

**Proposition 4.2.** *Let $E_i = E_i(\tilde{X})$, for $i = 1, \ldots, m$, with $mk \geq d$, denote the embeddings generated by the decoupler and regularized using HSIC, and let the concatenated embedding be $E(\tilde{X}) = \left[E_1(\tilde{X}); E_2(\tilde{X}); \ldots; E_m(\tilde{X})\right]$. Under exact HSIC regularization, we have $\text{rank}(E(\tilde{X})) = \sum_{i=1}^{m}\text{rank}(E_i(\tilde{X}))$. Moreover, under perturbation or approximate HSIC regularization, let the block covariance be $\Sigma_{\text{blk}} = \mathbb{E}\left[\text{vec}(E(\tilde{X}))\text{vec}(E(\tilde{X}))^\top\right]$, and let $\Sigma = \Sigma_{\text{blk}} + \Delta$, where $\Delta$ denotes the perturbation. If $\Sigma_{\text{blk}}$ has $r$ eigenvalues bounded below by $\gamma > 0$ and $\|\Delta\|_2 < \gamma$, then $\Sigma$ retains at least $r$ eigenvalues greater than $\gamma - \|\Delta\|_2 > 0$. Hence, the numerical rank of $\Sigma$ is preserved up to the spectral norm of the perturbation $\Delta$, implying that $\Sigma_{\text{blk}}$ and, consequently, $E(\tilde{X})$ maintains robustness of its spectrum.*

*Proof.* We separately consider the cases of exact HSIC-orthogonality and perturbation/approximate HSIC-orthogonality. For simplicity, we denote the input to each $E_i$ by $X$ (instead of $\tilde{X}$). Without loss of generality, assume that each $E_i(X)$ has finite second moments and that the embeddings are centered, i.e., $\mathbb{E}[E_i(X)] = 0$ for all $i$, which can be ensured via normalization techniques such as batch normalization (Ioffe & Szegedy, 2015).

### A.2.1 SPECTRUM ROBUSTNESS UNDER EXACT HSIC-ORTHOGONALITY

We define the population cross-covariance blocks as

$$C_{ij} = \mathbb{E}[\text{vec}(E_i)\,\text{vec}(E_j)^\top], \quad i \neq j.$$

Then, the block covariance of the concatenated embeddings, $\Sigma$, can be written as

$$\Sigma = \mathbb{E}[\text{vec}(E(X))\,\text{vec}(E(X))^\top] = \begin{bmatrix} \Sigma_1 & C_{12} & \cdots & C_{1m} \\ C_{21} & \Sigma_2 & \cdots & C_{2m} \\ \vdots & \vdots & \ddots & \vdots \\ C_{m1} & C_{m2} & \cdots & \Sigma_m \end{bmatrix}.$$

Under exact HSIC-orthogonality, where each $E_i$ is mutually independent, we have the following theorem on the rank additivity of the covariance $\Sigma$ and, consequently, of the concatenated embedding $E(X)$:

**Theorem A.1** (Rank additivity with exact HSIC-orthogonality). *Assume the kernels used for HSIC are characteristic and that for every pair $i \neq j$ we have*

$$\text{HSIC}(E_i, E_j) = 0$$

*in the population limit. Then the random matrices $E_1, \ldots, E_m$ are mutually independent; consequently the cross-covariance blocks vanish:*

$$C_{ij} = 0 \quad \text{for all } i \neq j.$$

*Hence the covariance $\Sigma$ is block-diagonal,*

$$\Sigma = \text{diag}(\Sigma_1, \ldots, \Sigma_m),$$

*and its rank satisfies*

$$\text{rank}(\Sigma) = \sum_{i=1}^m \text{rank}(\Sigma_i).$$

*That is, if each $\Sigma_i$ has nonzero rank, then $\Sigma$ cannot collapse to a rank smaller than the sum of the individual ranks. Consequently, when each $E_i$ is generated as a non-collapsing component, the concatenated embedding $E(X)$ preserves its rank as*

$$\text{rank}(E(X)) = \sum_{i=1}^m \text{rank}(E_i(X)).$$

*Proof.* Since the kernels are characteristic, $\text{HSIC}(E_i, E_j) = 0$ implies $E_i$ and $E_j$ are independent for every pair $i \neq j$. Mutual independence and centering ($\mathbb{E}[E_i] = 0$) imply, for $i \neq j$,

$$C_{ij} = \mathbb{E}[\text{vec}(E_i)\,\text{vec}(E_j)^\top] = 0.$$

Therefore $\Sigma$ is block-diagonal with diagonal blocks $\Sigma_1, \ldots, \Sigma_m$, and its rank follows the standard additivity property from linear algebra (Strang, 2022). Moreover, since

$$\text{rank}(X^\top X) = \text{rank}(X)$$

holds for any real matrix $X$, the concatenated embedding $E(X)$ preserves the same rank additivity across the individual $E_i$, thereby concluding the proof. $\square$

Theorem A.1 shows that under exact HSIC-orthogonality, the rank of the concatenated embedding $E(X)$ is fully determined by the independent components $E_i$ extracted by the embedding decoupler, exhibiting a rank additivity property rather than collapse. Since $mk \geq d$, the generated embedding sets are capable of preserving the full information, i.e., the matrix rank, of the input $X$, thereby preventing embedding collapse in the decoupler.

### A.2.2 Spectrum Robustness under Perturbation / Approximate HSIC

Before establishing spectrum (dimensional) robustness, we first present the following lemma on a matrix perturbation bound:

**Lemma A.2** (Weyl perturbation bound for symmetric PSD matrices)**.** *Let $A, B \in \mathbb{R}^{d \times d}$ be symmetric, with eigenvalues $\lambda_1(\cdot) \geq \lambda_2(\cdot) \geq \cdots \geq \lambda_d(\cdot)$. Then for every $j$,*

$$|\lambda_j(A) - \lambda_j(B)| \leq \|A - B\|_2,$$

*where $\| \cdot \|_2$ is the spectral norm.*

*Proof.* This is the classical Weyl's inequality; see any standard matrix analysis reference (Marcus & Minc, 1992). □

Next, we establish the spectrum (dimensional) robustness under perturbation or approximate HSIC regularization, summarized in the following theorem:

**Theorem A.3** (Rank preservation with perturbation / approximate HSIC)**.** *Write $\Sigma = \Sigma_{\mathrm{blk}} + \Delta$, where*

$$\Sigma_{\mathrm{blk}} = \mathrm{diag}(\Sigma_1, \ldots, \Sigma_m)$$

*is the block-diagonal covariance and $\Delta$ contains the off-diagonal cross-covariances (or modeling/estimation error). Let $\lambda_j(\cdot)$ denote the $j$-th largest eigenvalue. Then for every $j$,*

$$|\lambda_j(\Sigma) - \lambda_j(\Sigma_{\mathrm{blk}})| \leq \|\Delta\|_2.$$

*Consequently, if $\Sigma_{\mathrm{blk}}$ has $r$ eigenvalues lower-bounded by $\gamma > 0$ ($\lambda_r(\Sigma_{\mathrm{blk}}) \geq \gamma$), and $\|\Delta\|_2 < \gamma$, then $\Sigma$ has at least $r$ eigenvalues $\geq \gamma - \|\Delta\|_2 > 0$. In particular, the effective linear dimension (numerical rank) of $\Sigma$ is preserved up to the spectral-norm size of the cross-covariance perturbation $\Delta$.*

*Proof.* Apply Lemma A.2 with $A = \Sigma$ and $B = \Sigma_{\mathrm{blk}}$. The stated consequence about preserving at least $r$ eigenvalues above a positive threshold follows immediately. □

Therefore, if empirical HSIC penalties are used and (i) they are driven to small values for all pairs $(i, j)$, and (ii) small HSIC empirically implies a small cross-covariance spectral norm $\|\Delta\|_2$, then by Theorem A.3, the concatenated covariance $\Sigma$ preserves the high-rank directions present in the block-diagonal covariance $\Sigma_{\mathrm{blk}}$, and consequently the concatenated embedding $E(X)$ maintains its rank. Thus, HSIC regularization provides robustness against second-order dimensional collapse: the concatenated embedding resists collapsing into a low-dimensional linear subspace up to the magnitude of the residual cross-dependence.

□

### A.3 Proof of Proposition 4.3

**Proposition 4.3.** *Let $\theta_{j,i} > 0$ be the per-row weights generated by an MLP-parameterized field-wise gating mechanism. Then the gated embedding $Z = \sum_{i=1}^{m} D_i E_i$, with $D_i = \mathrm{diag}(\theta_{1,i}, \ldots, \theta_{n,i}) \in \mathbb{R}^{n \times n}$, which serves as the input to a feature interaction module, satisfies*

$$\max_i \mathrm{rank}(E_i) \ \leq \ \mathrm{rank}(Z) \ \leq \ \min\Big(k, \sum_{i=1}^{m} \mathrm{rank}(E_i)\Big).$$

*Consequently, $Z$ is guaranteed to be high-rank, ensuring that downstream feature interaction modules—such as FM (Rendle, 2010), DeepFM (Guo et al., 2017), DCN/CrossNet (Wang et al., 2021), and CIN (Lian et al., 2018)—operate on a non-collapsed embedding space.*

*Proof.* Since the embeddings $E_i$ for $i = 1, \ldots, m$ are optimized under the HSIC regularization, their components are linearly independent, implying that their column spaces form a direct sum:

$$\sum_{i=1}^{m} \mathrm{Col}(E_i) = \bigoplus_{i=1}^{m} \mathrm{Col}(E_i).$$

This implies that any vector in the sum of these spaces has a unique representation as a sum of vectors from each component space. A non-zero vector in one component's column space cannot be represented by a sum of vectors from the others.

Without loss of generality, we consider the input to a single feature interaction module. The feature interaction modules operate in parallel, each receiving distinct combinations of the component embeddings $E_i$. Specifically, the field-wise gating mechanism, parameterized by an MLP, produces per-row weights $\theta_{j,i} > 0$, yielding the gated embedding:

$$Z = \sum_{i=1}^{m} D_i E_i, \qquad D_i = \mathrm{diag}(\theta_{1,i}, \ldots, \theta_{n,i}) \in \mathbb{R}^{n \times n},$$

which serves as the input to a feature interaction module.

We can now establish a rank bound for such gated embeddings constructed via per-row gating, formalized in the following theorem:

**Theorem A.4** (Rank bounds under field-wise gating). *Let $r_i = \mathrm{rank}(E_i)$. Then the rank of the gated embedding satisfies*

$$\max_i r_i \ \leq \ \mathrm{rank}(Z) \ \leq \ \min\Big(k, \sum_{i=1}^{m} r_i\Big).$$

*Proof.* We prove the upper and lower bounds separately.

- **Upper bound:** The rank of a sum of matrices is at most the sum of their ranks (sub-additivity). Since each $D_i$ is a diagonal matrix with strictly positive entries, it is invertible and thus a full-rank linear transformation. Multiplying by $D_i$ does not alter the rank of the matrix, so $\mathrm{rank}(D_i E_i) = \mathrm{rank}(E_i) = r_i$.

$$\mathrm{rank}(Z) = \mathrm{rank}\Big(\sum_{i=1}^{m} D_i E_i\Big) \leq \sum_{i=1}^{m} \mathrm{rank}(D_i E_i) = \sum_{i=1}^{m} r_i.$$

Furthermore, since $Z \in \mathbb{R}^{n \times k}$, its rank cannot exceed $k$. Combining these gives the upper bound: $\mathrm{rank}(Z) \leq \min(k, \sum_i r_i)$.

- **Lower bound:** We will show that $\mathrm{rank}(Z) \geq r_i$ for any arbitrary component $i$, which implies the desired lower bound. Our strategy is to prove that the null space of $Z$ is a subspace of the null space of $E_i$ (i.e., $\mathrm{null}(Z) \subseteq \mathrm{null}(E_i)$).

Let $h \in \mathrm{null}(Z)$, which by definition means $Zh = 0$. Expanding $Z$, we have:

$$\sum_{i=1}^{m} D_i(E_i h) = 0.$$

Let us define a set of vectors $u_i = E_i h$, where each $u_i \in \mathrm{Col}(E_i)$. The equation becomes:

$$\sum_{i=1}^{m} D_i u_i = 0.$$

We hope to show this implies $u_i = 0$ for all $i$, which relies on the direct sum property being preserved after the transformation by the gating matrices $D_i$. Specifically, a linear dependency among the transformed spaces $\{\mathrm{Col}(D_i E_i)\}$ would require a non-generic, pathological alignment between the gating values in $\{D_i\}$ and the basis vectors of the spaces $\{\mathrm{Col}(E_i)\}$. Since the gating values $\theta_{j,i}$ are outputs of a trained MLP, they are not structured to create such specific cancellations. In any practical scenario, the gating matrices will not introduce new linear dependencies. Thus, the set of transformed spaces $\{\mathrm{Col}(D_i E_i)\}$ also forms a direct sum.

Since the spaces $\{\mathrm{Col}(D_i E_i)\}$ form a direct sum and we have a sum of vectors from these spaces equal to zero, each vector in the sum must be zero:

$$D_i u_i = 0 \quad \text{for all } i = 1, \ldots, m.$$

Because each $D_i$ is invertible (having strictly positive diagonal entries), $D_i u_i = 0$ implies $u_i = 0$. Therefore, it must be that $u_i = E_i h = 0$. This shows that any vector $h$ in the null space of $Z$ must also be in the null space of $E_i$, establishing that $\text{null}(Z) \subseteq \text{null}(E_i)$.

Since $\text{null}(Z)$ is a subspace of $\text{null}(E_i)$, its dimension must be less than or equal to that of $\text{null}(E_i)$:

$$\dim(\text{null}(Z)) \leq \dim(\text{null}(E_i)).$$

From the rank-nullity theorem ($\text{rank}(M) + \dim(\text{null}(M)) = k$), this directly implies:

$$\text{rank}(Z) \geq \text{rank}(E_i) = r_i.$$

As this holds for any component $i$, the rank of $Z$ must be at least the maximum of all individual ranks, concluding the proof.

$\square$

Based on the theoretical guarantees of Theorem A.4, we can now analyze how our design alleviates rank collapse in feature interaction across several representative feature interaction models. Specifically, let $r = \text{rank}(Z)$, and denote the rows of $Z$ by $z_j^\top$.

**Corollary A.5** (FM). *The FM Gram matrix $G = ZZ^\top$ satisfies $\text{rank}(G) = r$. A higher rank $r$ increases the expressive power of pairwise interactions, directly combating rank collapse.*

**Corollary A.6** (DeepFM (DNN branch)). *Let the DNN's first linear layer be $W \in \mathbb{R}^{h \times (nk)}$. Across samples, the input matrix $F$ (flattened $Z$) is drawn from a subspace whose dimension scales with $r$. Thus,*

$$\text{rank}(WF) \leq \min(\text{rank}(W), nr).$$

*A larger $r$ allows a richer and higher-dimensional input space for the DNN branch to learn from.*

**Corollary A.7** (DCN / CrossNet). *Let the pooled vector $g_0$ be a linear map of the rows of $Z$. Each cross-layer update*

$$g^{(l+1)} = g_0(w_l^\top g^{(l)}) + g^{(l)} + b_l$$

*is confined to the subspace spanned by $g_0$. The dimensionality of the space where $g_0$ can exist is bounded by $r$. Increasing $r$ enlarges this reachable subspace, allowing for more complex cross-features.*

**Corollary A.8** (CIN). *The first-layer feature map in CIN consists of pairwise element-wise products $\{z_i \odot z_j\}$. If the vectors $\{z_i\}$ span an $r$-dimensional space, the resulting product vectors will span a space of dimension up to $O(r^2)$. Thus, CIN's representational capacity grows super-linearly with $r$, making a high-rank $Z$ highly beneficial.*

Based on the corollaries above, our field-wise gating applied to independent embedding sets effectively mitigates interaction collapse. Each component $E_i$ provides an independent "view" of the feature fields, and per-row gating constructs each field embedding as a weighted mixture of these views. Under standard conditions, this mixture preserves the linear independence of the components, ensuring that the resulting gated embedding $Z$ is high-rank. Since the rank of $Z$ is strongly correlated with the rank of outputs in downstream feature interaction modules, the high-rank embeddings generated by our method directly prevent interaction collapse in modules such as FM, DeepFM, DCN/CrossNet, and CIN.

$\square$

## A.4 Experimental Details

### A.4.1 Dataset Description

The Criteo and Avazu datasets used in our experiments are sourced from the open-source Python library FuxiCTR (Zhu et al., 2021). Specifically, we use the Criteo x1 and Avazu x4 datasets, and their statistics are summarized in Table 4.

Table 4: Statistics of benchmark datasets used in our experiments.

| Dataset | #Train Size | #Valid Size | #Test Size | #Fields |
|---------|-------------|-------------|------------|---------|
| Criteo  | 33.0M       | 8.3M        | 4.6M       | 39      |
| Avazu   | 32.3M       | 4.0M        | 4.0M       | 24      |

### A.4.2 EXPERIMENTAL SETTINGS

The experiments are conducted on Ubuntu 22.04 with Python 3.10, using an NVIDIA A100 GPU (40GB) with CUDA 11.8.

For all baselines' multi-embedding variants, we follow the strategy in (Guo et al., 2024b), parallelizing multiple embedding tables at the base size, with the number of tables determined by the scale-up factor. For RoBlock, the scale-up factor determines $m$, the number of independent components.

For all shallow architecture baselines and Wukong, we directly adopt the implementations provided by FuxiCTR (Zhu et al., 2021). For DHEN, the gating module is implemented as a 3-layer MLP with 64 hidden units, and its supported interaction functions include FM (Rendle, 2010), CIN (Lian et al., 2018), DNN, and DCNv2 (Wang et al., 2021). These interaction modules follow the FuxiCTR implementations, and their combinations in the multi-embedding setting are generated via sampling without replacement across cycles, controlled by the random seed to promote diversity. For RoBlock, the base variant adopts DCNv2 (CrossNetv2) as the interaction module, while the scaled-size variants employ the same set of interaction functions as DHEN. All MLP components are standardized to three layers; in particular, the gating MLP employs 64 hidden units, matching the configuration used in DHEN. Rank-1 update normalization is performed with three iteration steps, following prior empirical practice (Chen et al., 2022; Wang et al., 2024; Yin et al., 2024).

Due to differences between our experimental setup and that of FuxiCTR, we fine-tune only the **model-specific** hyperparameters (e.g., number of MLP layers, hidden dimensions, interaction orders, activation functions). The associated search spaces are expanded based on the hyperparameter configurations reported for each model in FuxiCTR.

In contrast, to ensure controlled and fair comparisons across models, we **unify all universal hyperparameters** within the FuxiCTR training framework. Specifically, we set the learning rate to 0.001, embedding regularization to 1e–8, embedding dropout to 0, network regularization to 0, network dropout to 0.1, stop patience to 2, the base embedding size to 10, and the batch size to 10,000. Model optimization is performed with the Adam optimizer (Kingma & Ba, 2017) using binary cross-entropy loss, following the FuxiCTR configuration. Additionally, RoBlock incorporates an auxiliary HSIC regularization loss, $\mathcal{L}_{\text{HSIC}}$, with a tunable trade-off coefficient $\beta \in \{0, 0.1, 0.5, 1, 10\}$. Experimental results are averaged over 20 random initializations to ensure statistical robustness. To measure the information abundance (IA) of the embeddings, we compute the IA of the layer inputs for vanilla DHEN, vanilla Wukong, and our RoBlock. For the multi-embedding variants of DHEN and Wukong, we first concatenate the multiple embedding sets before calculating the IA at each depth.

### A.4.3 COMPLEMENTARY RESULTS

**Complete results with standard deviations.** In main text of Tables 2, 1, and 3, we only report average test AUC due to space limit. Here we present detailed experimental results with estimated standard deviation for statistical rigor, with results shown in Tables 5, 6, 7, and 8.

**Full results for scaling analysis of RoBlock in depth and width.** To comprehensively investigate the scaling behavior of RoBlock under different embedding size settings, we extend the results in Figure 6 by providing additional analyses across a broader range of embedding scale factors, as discussed in Section 5.4.

As shown in these figures, for each scale factor ranging from a moderate 2x scale-up to a large 10x scale-up, increasing both the model depth and width consistently improves task performance. Notably, the combined effect of jointly scaling depth and width yields more substantial gains than scaling either dimension independently. More specifically, these results offer a decoupled analysis of the number of components $m$ and the embedding size $d$. Even when the two are not tied by the

Table 5: Test AUC of shallow models on Criteo with mean $\pm$ standard deviation. These values complement the mean results in Table 1.

| Model | | Criteo | | | | |
| --- | --- | --- | --- | --- | --- | --- |
| | | base | 2× | 3× | 5× | 10× |
| DNN | SE | $0.81092_{\pm 0.00012}$ | $0.81073_{\pm 0.00015}$ | $0.81092_{\pm 0.00016}$ | $0.81062_{\pm 0.00021}$ | $0.81049_{\pm 0.00022}$ |
| | ME | | $0.81125_{\pm 0.00017}$ | $0.81138_{\pm 0.00018}$ | $0.81150_{\pm 0.00019}$ | $0.81168_{\pm 0.00021}$ |
| Crossnet | SE | $0.81162_{\pm 0.00013}$ | $0.81163_{\pm 0.00009}$ | $0.81140_{\pm 0.00011}$ | $0.81152_{\pm 0.00014}$ | $0.81124_{\pm 0.00017}$ |
| | ME | | $0.81190_{\pm 0.00011}$ | $0.81208_{\pm 0.00013}$ | $0.81222_{\pm 0.00016}$ | $0.81245_{\pm 0.00019}$ |
| NFM | SE | $0.80862_{\pm 0.00014}$ | $0.80852_{\pm 0.00009}$ | $0.80862_{\pm 0.00016}$ | $0.80847_{\pm 0.00016}$ | $0.80824_{\pm 0.00019}$ |
| | ME | | $0.80884_{\pm 0.00009}$ | $0.80904_{\pm 0.00013}$ | $0.80915_{\pm 0.00018}$ | $0.80933_{\pm 0.00022}$ |
| CIN | SE | $0.81171_{\pm 0.00014}$ | $0.81168_{\pm 0.00011}$ | $0.81171_{\pm 0.00012}$ | $0.81170_{\pm 0.00013}$ | $0.81146_{\pm 0.00018}$ |
| | ME | | $0.81199_{\pm 0.00009}$ | $0.81218_{\pm 0.00010}$ | $0.81226_{\pm 0.00012}$ | $0.81248_{\pm 0.00020}$ |
| xDeepFM | SE | $0.81348_{\pm 0.00020}$ | $0.81340_{\pm 0.00014}$ | $0.81322_{\pm 0.00015}$ | $0.81333_{\pm 0.00019}$ | $0.81298_{\pm 0.00019}$ |
| | ME | | $0.81366_{\pm 0.00013}$ | $0.81375_{\pm 0.00013}$ | $0.81392_{\pm 0.00014}$ | $0.81421_{\pm 0.00017}$ |
| DCNv2 | SE | $0.81315_{\pm 0.00009}$ | $0.81304_{\pm 0.00010}$ | $0.81315_{\pm 0.00017}$ | $0.81300_{\pm 0.00022}$ | $0.81276_{\pm 0.00022}$ |
| | ME | | $0.81336_{\pm 0.00010}$ | $0.81348_{\pm 0.00011}$ | $0.81362_{\pm 0.00016}$ | $0.81395_{\pm 0.00016}$ |
| RoBlock | | $0.81382_{\pm 0.00011}$ | $0.81401_{\pm 0.00010}$ | $0.81413_{\pm 0.00011}$ | $0.81424_{\pm 0.00013}$ | $0.81464_{\pm 0.00017}$ |

Table 6: Test AUC of shallow models on Avazu with mean $\pm$ standard deviation. These values complement the mean results in Table 2.

| Model | | Avazu | | | | |
| --- | --- | --- | --- | --- | --- | --- |
| | | base | 2× | 3× | 5× | 10× |
| DNN | SE | $0.78922_{\pm 0.00011}$ | $0.78920_{\pm 0.00010}$ | $0.78904_{\pm 0.00011}$ | $0.78914_{\pm 0.00011}$ | $0.78896_{\pm 0.00022}$ |
| | ME | | $0.78958_{\pm 0.00011}$ | $0.78966_{\pm 0.00012}$ | $0.78982_{\pm 0.00015}$ | $0.79004_{\pm 0.00020}$ |
| Crossnet | SE | $0.79010_{\pm 0.00013}$ | $0.79004_{\pm 0.00012}$ | $0.78995_{\pm 0.00012}$ | $0.79002_{\pm 0.00013}$ | $0.78983_{\pm 0.00013}$ |
| | ME | | $0.79038_{\pm 0.00015}$ | $0.79045_{\pm 0.00016}$ | $0.79061_{\pm 0.00017}$ | $0.79086_{\pm 0.00020}$ |
| NFM | SE | $0.78799_{\pm 0.00010}$ | $0.78796_{\pm 0.00010}$ | $0.78802_{\pm 0.00013}$ | $0.78790_{\pm 0.00013}$ | $0.78764_{\pm 0.00016}$ |
| | ME | | $0.78828_{\pm 0.00012}$ | $0.78846_{\pm 0.00013}$ | $0.78856_{\pm 0.00018}$ | $0.78891_{\pm 0.00018}$ |
| CIN | SE | $0.79211_{\pm 0.00010}$ | $0.79207_{\pm 0.00015}$ | $0.79218_{\pm 0.00017}$ | $0.79212_{\pm 0.00018}$ | $0.79192_{\pm 0.00020}$ |
| | ME | | $0.79237_{\pm 0.00008}$ | $0.79251_{\pm 0.00009}$ | $0.79265_{\pm 0.00010}$ | $0.79290_{\pm 0.00019}$ |
| xDeepFM | SE | $0.79343_{\pm 0.00014}$ | $0.79336_{\pm 0.00009}$ | $0.79342_{\pm 0.00016}$ | $0.79330_{\pm 0.00019}$ | $0.79302_{\pm 0.00020}$ |
| | ME | | $0.79369_{\pm 0.00015}$ | $0.79383_{\pm 0.00016}$ | $0.79396_{\pm 0.00018}$ | $0.79420_{\pm 0.00022}$ |
| DCNv2 | SE | $0.79403_{\pm 0.00009}$ | $0.79404_{\pm 0.00008}$ | $0.79411_{\pm 0.00008}$ | $0.79413_{\pm 0.00012}$ | $0.79405_{\pm 0.00013}$ |
| | ME | | $0.79417_{\pm 0.00009}$ | $0.79427_{\pm 0.00012}$ | $0.79436_{\pm 0.00018}$ | $0.79452_{\pm 0.00021}$ |
| RoBlock | | $0.79493_{\pm 0.00013}$ | $0.79543_{\pm 0.00009}$ | $0.79611_{\pm 0.00011}$ | $0.79650_{\pm 0.00013}$ | $0.79683_{\pm 0.00017}$ |

default relation $m = \lfloor d/k \rfloor + 1$ but are instead flexibly configured, RoBlock continues to deliver consistent performance gains with increased model capacity. This highlights the flexible and scalable nature of RoBlock, showcasing its ability to adjust model capacity across multiple dimensions to meet the demands of complex real-world scenarios.

**Additional Experiments on Heterogeneous Feature Interaction.** Although the effectiveness of heterogeneous feature interaction modules has been empirically demonstrated (Zeng et al., 2025; Zhang et al., 2022; Wang et al., 2025b) and is **not the main focus / contribution of this paper**, we include complementary evaluations to provide additional insights for readers. Specifically, we conduct ablation studies comparing heterogeneous and homogeneous feature interaction modules. Using a 5-layer RoBlock as the backbone, we randomly construct 10 different combinations of feature interaction modules. In the homogeneous setting, all modules in a combination are of the same type, whereas in the heterogeneous setting, not all modules are identical. We report the average test AUC across the 10 combinations; complete results are provided in Table 9.

As shown, using heterogeneous feature interaction modules leads to statistically stronger performance. Moreover, the performance gain from increasing embedding size is slightly higher for heterogeneous modules compared to homogeneous ones. This observation aligns with recent findings, indicating that RoBlock can also benefit from heterogeneous feature interactions.

Table 7: Test AUC of deep models on Avazu with mean ± standard deviation. These values complement the mean results in Table 3.

| Model | #layers | Criteo | | | | |
|---|---|---|---|---|---|---|
| | | base | 2× | 3× | 5× | 10× |
| DHEN | 2 | 0.813457 | 0.813584 | 0.813714 | 0.813873 | 0.814207 |
| | | ±0.000284 | ±0.000115 | ±0.000157 | ±0.000206 | ±0.000268 |
| | 3 | 0.813556 | 0.813661 | 0.813782 | 0.813961 | 0.814280 |
| | | ±0.000185 | ±0.000192 | ±0.000213 | ±0.000259 | ±0.000305 |
| | 4 | 0.813597 | 0.813720 | 0.813834 | 0.814009 | 0.814315 |
| | | ±0.000207 | ±0.000237 | ±0.000254 | ±0.000304 | ±0.000246 |
| | 5 | 0.813641 | 0.813789 | 0.813870 | 0.814057 | 0.814349 |
| | | ±0.000178 | ±0.000261 | ±0.000213 | ±0.000238 | ±0.000302 |
| Wukong | 2 | 0.813316 | 0.813427 | 0.813484 | 0.813631 | 0.813680 |
| | | ±0.000251 | ±0.00019 | ±0.000213 | ±0.000275 | ±0.000252 |
| | 3 | 0.813394 | 0.813520 | 0.813571 | 0.813721 | 0.813781 |
| | | ±0.000163 | ±0.000198 | ±0.000221 | ±0.000253 | ±0.000311 |
| | 4 | 0.813425 | 0.813579 | 0.813626 | 0.813757 | 0.813816 |
| | | ±0.000187 | ±0.000307 | ±0.000317 | ±0.000320 | ±0.000310 |
| | 5 | 0.813475 | 0.813632 | 0.813683 | 0.813804 | 0.813859 |
| | | ±0.000167 | ±0.000230 | ±0.000252 | ±0.000273 | ±0.000310 |
| RoBlock | 2 | 0.813859 | 0.814065 | 0.814165 | 0.814279 | 0.814689 |
| | | ±0.000121 | ±0.000222 | ±0.000224 | ±0.000247 | ±0.000280 |
| | 3 | 0.813968 | 0.814148 | 0.814254 | 0.814363 | 0.814722 |
| | | ±0.000174 | ±0.000206 | ±0.000234 | ±0.000268 | ±0.000217 |
| | 4 | 0.814016 | 0.814182 | 0.814293 | 0.814404 | 0.814765 |
| | | ±0.000152 | ±0.000133 | ±0.000266 | ±0.000245 | ±0.000266 |
| | 5 | 0.814070 | 0.814258 | 0.814357 | 0.814480 | 0.814802 |
| | | ±0.000196 | ±0.000235 | ±0.000242 | ±0.000320 | ±0.000233 |

### A.4.4 COMPUTATIONAL AND MEMORY COMPLEXITY ANALYSIS OF RANK-1 UPDATES

Let $X \in \mathbb{R}^{b \times n \times d}$ be a **batch** of input embeddings, and $n$ and $d$ are the dimensions of each matrix. Our rank-1 update normalization performs $t$ iterations for each batch sample.

- **Time Complexity.** Taking out the upper notions for $v$, each iteration requires computing $v \leftarrow X^\top(Xv)$ for every batch sample. First, we compute $Xv$, which has a cost of $\mathcal{O}(bnd)$, and then we compute $X^\top(Xv)$, which also costs $\mathcal{O}(bnd)$. Over $t$ iterations, the total time complexity is therefore $\mathcal{O}(tbnd)$. Computing the rank-1 approximation $X' = \frac{Xvv^\top}{\|v\|_2^2}$ also costs $\mathcal{O}(bnd)$, so the overall time complexity remains $\mathcal{O}(tbnd)$.

- **Space Complexity.** The main memory usage comes from storing the intermediate vectors $v \in \mathbb{R}^{b \times d}$, the intermediate matrix $Xv \in \mathbb{R}^{b \times n}$, and the rank-1 update $X' \in \mathbb{R}^{b \times n \times d}$. Therefore, the total space complexity is dominated by $\mathcal{O}(bnd)$, which scales linearly with the batch size and matrix dimensions.

However, note that $n$, the number of fields, is typically small (see Appendix A.4.1), and $t$, the number of iterations, is negligible in line with prior rank-1 update practices Chen et al. (2022); Wang et al. (2024); Yin et al. (2024). As a result, the overall complexity depends primarily on $b \times d$, i.e., the batch size and embedding dimension, which are easily adjustable, allowing the method to remain highly efficient.

Table 8: Test AUC of deep models on Avazu with mean $\pm$ standard deviation. These values complement the mean results in Table 3.

| Model | #layers | Avazu | | | | |
|---|---|---|---|---|---|---|
| | | base | 2× | 3× | 5× | 10× |
| DHEN | 2 | 0.793945 | 0.794096 | 0.794251 | 0.794396 | 0.794611 |
| | | ±0.000201 | ±0.000262 | ±0.000207 | ±0.000248 | ±0.000219 |
| | 3 | 0.794054 | 0.794186 | 0.794343 | 0.794471 | 0.794692 |
| | | ±0.000171 | ±0.000201 | ±0.000244 | ±0.000257 | ±0.000298 |
| | 4 | 0.794089 | 0.794218 | 0.794399 | 0.794519 | 0.794743 |
| | | ±0.000181 | ±0.000281 | ±0.000310 | ±0.000302 | ±0.000174 |
| | 5 | 0.794128 | 0.794278 | 0.794446 | 0.794556 | 0.794789 |
| | | ±0.000189 | ±0.000242 | ±0.000265 | ±0.000259 | ±0.000234 |
| Wukong | 2 | 0.793710 | 0.793822 | 0.793904 | 0.793980 | 0.794100 |
| | | ±0.000184 | ±0.000192 | ±0.000210 | ±0.000276 | ±0.000249 |
| | 3 | 0.793776 | 0.793903 | 0.794009 | 0.794078 | 0.794194 |
| | | ±0.000155 | ±0.000214 | ±0.000236 | ±0.000281 | ±0.000201 |
| | 4 | 0.793812 | 0.793942 | 0.794055 | 0.794121 | 0.794233 |
| | | ±0.000172 | ±0.000168 | ±0.000263 | ±0.000213 | ±0.000222 |
| | 5 | 0.793866 | 0.793985 | 0.794091 | 0.794184 | 0.794293 |
| | | ±0.000218 | ±0.000254 | ±0.000225 | ±0.000237 | ±0.000311 |
| RoBlock | 2 | 0.794981 | 0.795474 | 0.796155 | 0.796542 | 0.796865 |
| | | ±0.000173 | ±0.000172 | ±0.000158 | ±0.000233 | ±0.000214 |
| | 3 | 0.795075 | 0.795560 | 0.796263 | 0.796619 | 0.796928 |
| | | ±0.000161 | ±0.000218 | ±0.000251 | ±0.000276 | ±0.000259 |
| | 4 | 0.795129 | 0.795596 | 0.796308 | 0.796667 | 0.796959 |
| | | ±0.000211 | ±0.000248 | ±0.000278 | ±0.000227 | ±0.000269 |
| | 5 | 0.795208 | 0.795657 | 0.796364 | 0.796725 | 0.797022 |
| | | ±0.000217 | ±0.000260 | ±0.000232 | ±0.000206 | ±0.000132 |

Table 9: Ablation study on heterogeneous and homogeneous feature interaction configuration.

| Model | dataset | 2× | 3× | 5× | 10× |
|---|---|---|---|---|---|
| RoBlock$_{Hetero}$ | Criteo | **0.814258** | **0.814357** | **0.814480** | **0.814802** |
| | | ±0.000235 | ±0.000242 | ±0.000320 | ±0.000233 |
| | Avazu | **0.795657** | **0.796364** | **0.796725** | **0.797022** |
| | | ±0.000260 | ±0.000232 | ±0.000206 | ±0.000132 |
| RoBlock$_{Homo}$ | Criteo | 0.813831 | 0.813962 | 0.814203 | 0.814492 |
| | | ±0.000218 | ±0.000195 | ±0.000221 | ±0.000231 |
| | Avazu | 0.794722 | 0.794911 | 0.795265 | 0.795617 |
| | | ±0.000224 | ±0.000219 | ±0.000207 | ±0.000282 |

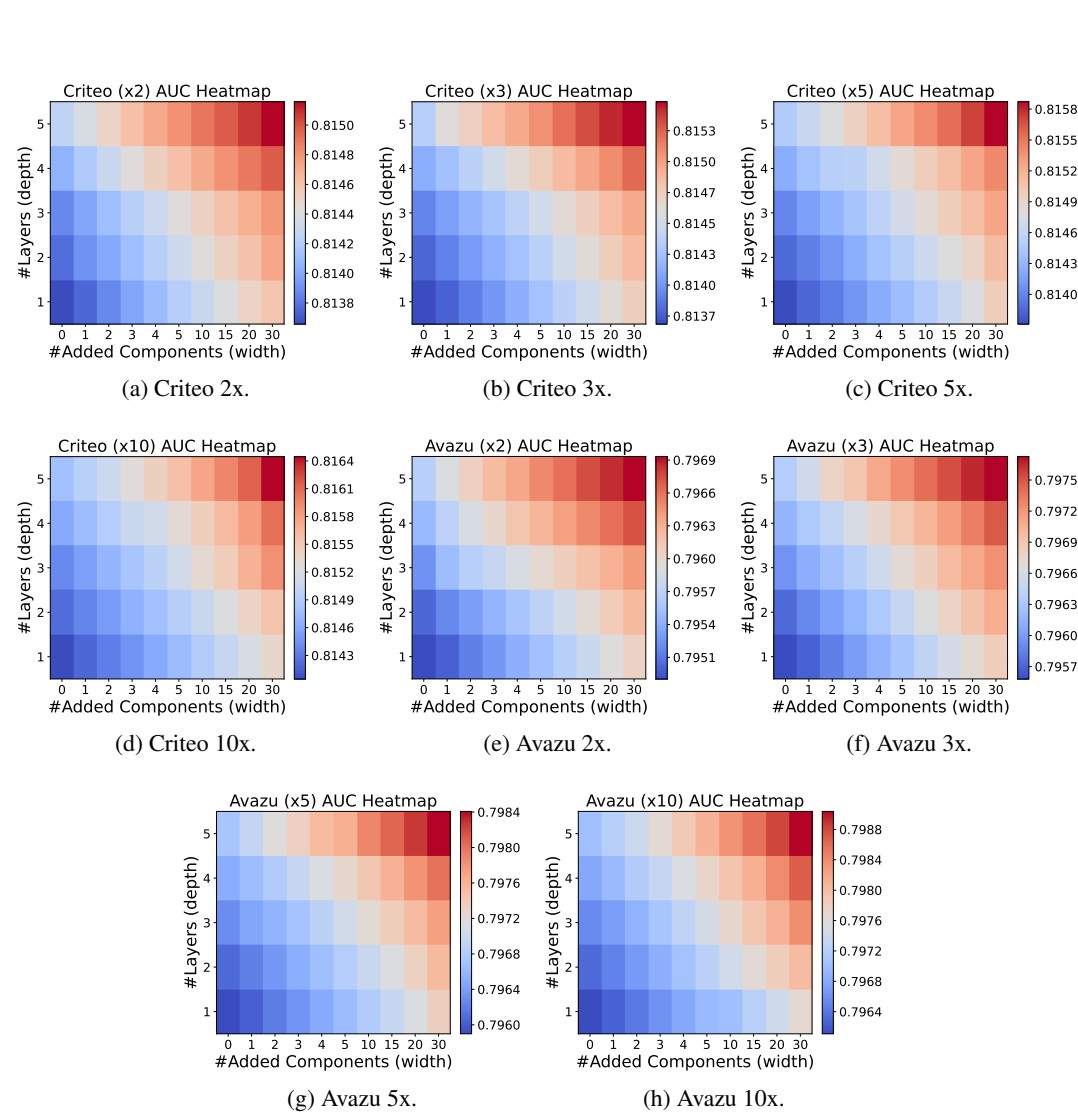

Figure 8: Complete RoBlock test AUC across varying numbers of layers and model scales relative to the base size. These results complement the mean values shown in Figure 6.

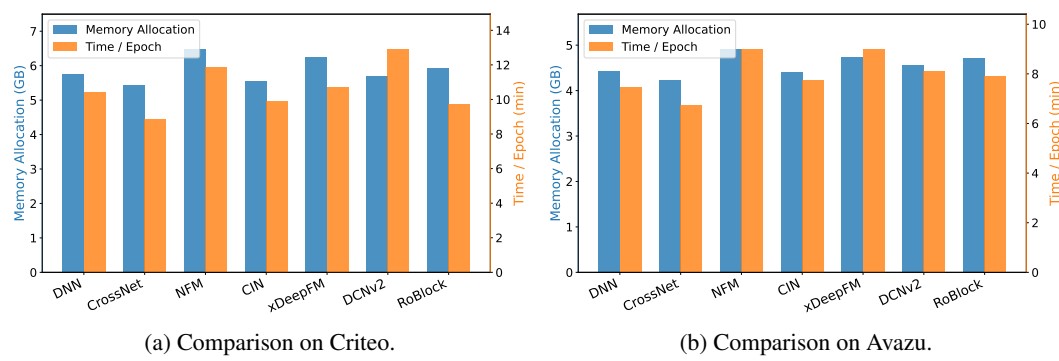

(a) Comparison on Criteo.         (b) Comparison on Avazu.

Figure 9: GPU memory allocation and training time of shallow baselines and RoBlock. All models are evaluated under a 10× scale-up.

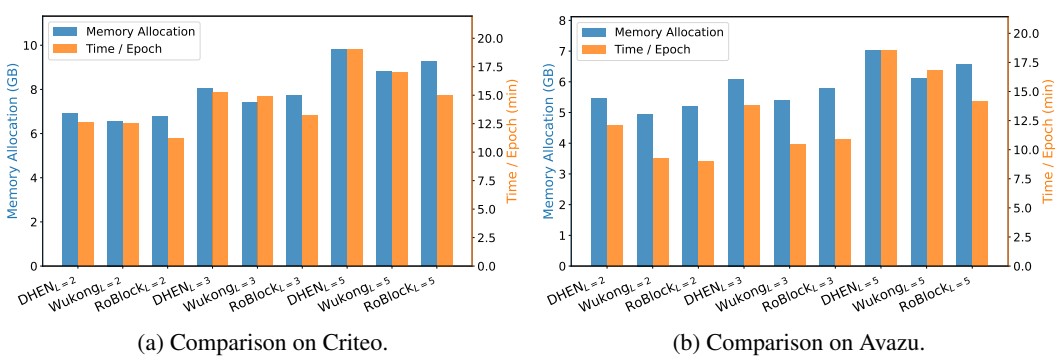

(a) Comparison on Criteo.         (b) Comparison on Avazu.

Figure 10: GPU memory allocation and training time of deep baselines and RoBlock. All models are evaluated under a 10× scale-up.

## A.5 ADDITIONAL EXPERIMENTS AND DISCUSSIONS IN REBUTTAL

In this part of the appendix, we provide additional results and discussions addressing the questions raised by the reviewers. These results are planned to be incorporated into the main text in the camera-ready version. Specifically, we cover *(i) efficiency evaluations*, *(ii) complexity of the HSIC loss*, and *(iii) additional ablation studies on $\beta$*.

### A.5.1 COMPUTATIONAL EFFICIENCY EVALUATIONS

To evaluate the efficiency of RoBlock, we examine *(i) GPU memory allocation*, *(ii) training time per epoch*, and *(iii) inference latency per batch*, and provide a comprehensive analysis of the results.

**Comparison on GPU Memory Allocation.** To facilitate an intuitive efficiency comparison, we evaluate all models using the largest 10× scale-up (i.e., embedding dimension of 100). For fairness, when comparing with shallow baselines, we evaluate RoBlock in its 1-layer configuration as used in the main text; similarly, when comparing with deep architectures, we benchmark baselines and our RoBlock across multiple layers to ensure comprehensive assessments. All baselines adopt the multi-embedding variants to remain consistent with the experimental setup in the main text.

As shown in the **blue** segments of Figures 9 and 10, RoBlock demonstrates **leading GPU memory efficiency** relative to the baselines, remaining comparable to both shallow and deep counterparts. This efficiency indicates that neither the rank-1 update nor the HSIC regularization introduces disproportionate overhead during training, consistent with the analysis in Appendix A.4.4 and A.5.2.

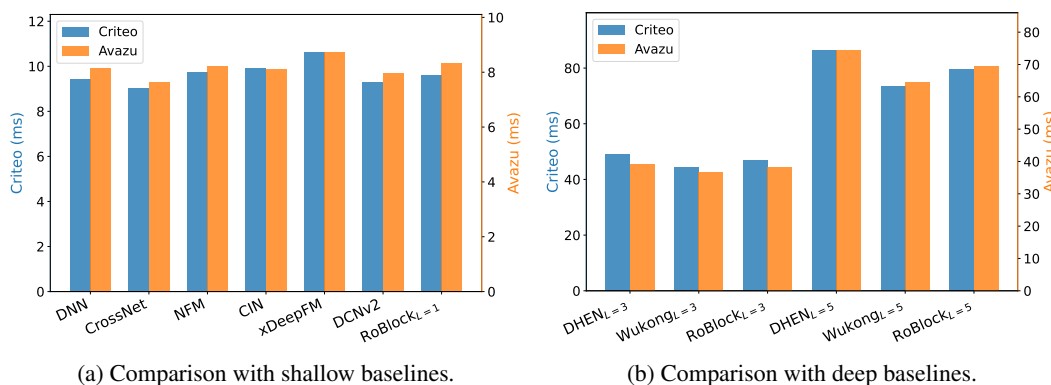

(a) Comparison with shallow baselines.

(b) Comparison with deep baselines.

Figure 11: Inference latency per batch of baselines and RoBlock. All models are evaluated under a 10× scale-up.

Quantitatively, the 1-layer RoBlock ranks *fourth lowest* among shallow baselines on both datasets, reducing GPU memory usage by up to $5\%$ compared with the strongest AUC baseline, xDeepFM, while incurring less than a $4\%$ increase relative to DCNv2. Multi-layer variants further rank *second lowest* among deep baselines, achieving up to a $6\%$ reduction compared with the strongest deep baseline, DHEN. These results reinforce that RoBlock provides enhanced modeling capacity without sacrificing practical efficiency.

**Comparison on Training Time Complexity.** We further evaluate training efficiency under the same settings as the memory comparison, reporting the average training time per epoch in Figures 9 and 10. We adopt per-epoch timing instead of total training time because the latter is highly sensitive to random initialization: different seeds may lead to substantially different numbers of training epochs, resulting in unstable and ambiguous comparisons. In contrast, the average time per epoch is more stable and therefore more informative.

As shown by the orange segments of Figures 9 and 10, in line with the GPU memory results, RoBlock achieves **leading training time efficiency**, matching or exceeding the performance of the baseline models. This efficiency primarily arises from the predominance of lightweight MLP modules, the low computational cost of both the HSIC loss and the rank-1 update, and the simplicity of RoBlock's interaction functions, which directly inherit efficient components from prior work, such as using only the CrossNetv2 module from DCNv2.

Specifically, the 1-layer RoBlock ranks as the *second fastest* on Criteo and *third fastest* on Avazu, achieving up to a $24\%$ speedup over DCNv2 on Criteo and up to a $13\%$ speedup over xDeepFM on Avazu. The multi-layer variants achieve the *fastest training speed* among deep baselines, reducing training time by up to $20\%$ compared with DHEN. These results confirm that RoBlock combines strong modeling capacity with top-tier training efficiency.

**Comparison on Inference Latency.** We also compare inference efficiency by measuring per-batch inference latency, using the same batch size of 10,000 as in training. The average latency per batch is reported in Figure 11.

We observe that inference latency patterns remain consistent within both shallow and deep model families, with RoBlock variants exhibiting latency comparable to their respective baselines. Since the HSIC regularization and rank-1 update are applied only during training and omitted at inference, RoBlock introduces no additional overhead. As a result, it maintains **competitive inference efficiency**, with latency staying within $2\%$ of the fastest baselines across both shallow and deep model groups, while delivering stronger predictive performance.

A.5.2    COMPLEXITY OF HSIC REGULARIZATION

In our implementation of HSIC regularization, we adopt an equivalent but more direct and efficient formulation. Specifically, leveraging standard results from matrix analysis Marcus & Minc (1992), we use $\mathrm{tr}(ABC) = \mathrm{tr}(BCA)$ and $\mathrm{tr}(AA^\top) = \|A\|_F^2$. Additionally, the centering matrix $H = I - \frac{1}{n}\mathbf{1}\mathbf{1}^\top$ is idempotent, satisfying $H \cdot H = H$. Using these properties, the trace computation of the linear-kernel HSIC (following the notations in Eq. 3) can be rewritten as

$$
\begin{aligned}
\mathrm{tr}\left(K(E_p^{(l-1)})HL(E_q^{(l-1)})H\right) &= \mathrm{tr}\left(E_p^{(l-1)}(E_p^{(l-1)})^\top (H \cdot H)E_q^{(l-1)}(E_q^{(l-1)})^\top (H \cdot H)\right), \\
&= \mathrm{tr}\left((HE_p^{(l-1)}(E_p^{(l-1)})^\top H) \cdot (HE_q^{(l-1)}(E_q^{(l-1)})^\top H)\right), \\
&= \left\|\left(HE_p^{(l-1)}\right)^\top \left(HE_q^{(l-1)}\right)\right\|_F^2.
\end{aligned}
\tag{15}
$$

This formulation not only simplifies computation but also significantly improves efficiency. The **memory complexity** of HSIC regularization is $\mathcal{O}(bnk + bk^2m^2)$. Since the field number $n$, base size $k$, and number of components $m$ are typically much smaller than the batch size $b$, the complexity is comppletely dominated by the batch. where the practical memory overhead is controllable in a reasonable batch size. Similarly, the **time complexity** is $\mathcal{O}(bnk^2m^2)$, which also can be controlled in a practically usable training time.

In summary, this efficient implementation of HSIC regularization provides meaningful performance gains for RoBlock while requiring reasonable additional computational resources, making it highly practical for training.

### A.5.3   ADDITIONAL ABLATIONS ON THE TRADE-OFF $\beta$

To further validate the effectiveness of our HSIC regularization, we provide additional ablation studies on the trade-off coefficient $\beta$. Specifically, using a 5-layer RoBlock as the backbone, we evaluate $\beta$ over a wide range of values $\{0, 0.1, 0.5, 1, 10, 100\}$. Experiments are conducted on both datasets and across multiple scale-up factors, offering extensive assessment of the impact of $\beta$.

As shown in Figure 12, when $\beta = 0$ (i.e., without HSIC regularization), the model exhibits a clear performance drop, confirming the effectiveness of incorporating HSIC regularization. The best performance is generally achieved with $\beta = 1$, and occasionally with 0.5 or 10, while extremely small (0.1) or large (100) values lead to noticeable degradation. These results indicate that RoBlock is not overly sensitive to $\beta$, and that a simple setting of $\beta = 1$ provides a robust and effective choice in practice.

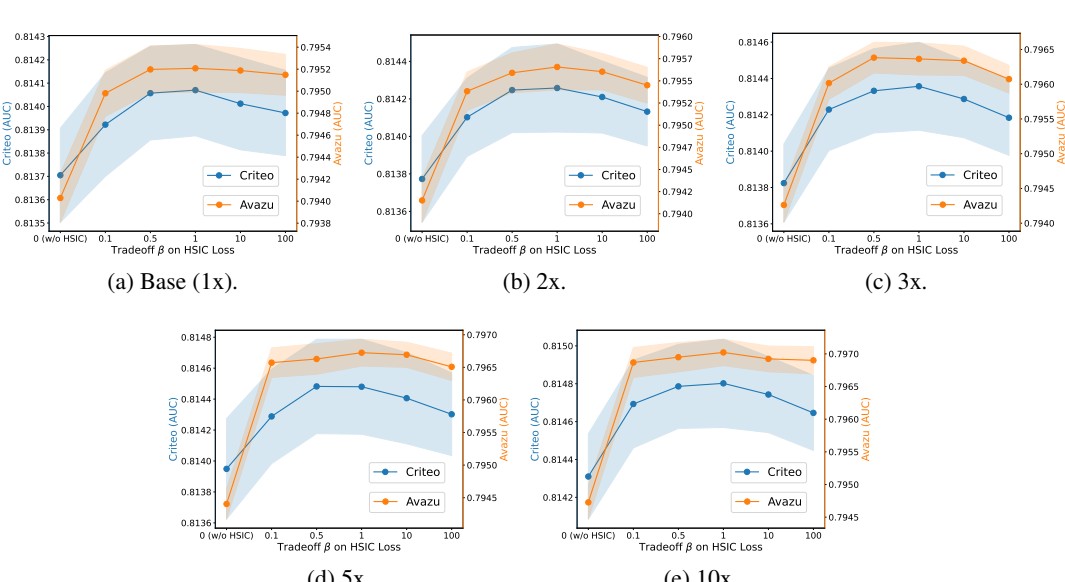

Figure 12: Impact of the trade-off coefficient $\beta$ under different scale-up factors.

