# OpenReview forum: "RoBlock: Wide and Deep Scaling of Recommenders via Embedding Collapse Mitigation"
_ICLR.cc/2026/Conference — Submitted to ICLR 2026_

### Official Review · Reviewer_mbei · 2025-10-21

**Soundness:** 3
**Presentation:** 3
**Contribution:** 3
**Rating:** 6
**Confidence:** 3

**Summary:**

This paper tackles the problem of embedding rank collapse in large-scale recommendation models, which manifests both in width (multi-embedding) and depth (layer stacking).
Existing works mainly address width-wise collapse (e.g., through multi-embedding fusion), while ignoring depth-wise degradation that occurs as the network deepens.
The authors propose RoBlock, a stackable modular block that mitigates rank collapse through three integrated mechanisms:
Rank-1 Update Normalization – balances the singular value spectrum of embeddings to prevent dominant directions.


HSIC-based Decoupler – enforces independence across embedding subspaces via Hilbert–Schmidt Independence Criterion(HSIC).


Field-wise Router – recombines embeddings across fields to regenerate non-collapsed components.


Theoretical analysis rigorously proves that these modules together preserve effective rank under both exact and approximate HSIC orthogonality. Empirical studies on Criteo and Avazu datasets show consistent improvements in AUC and information abundance (IA) across depth and width scaling.

**Strengths:**

**Design Strength**

- **Logical three-stage pipeline**: Clear progression from Stabilize (Rank-1 Normalization) → Decorrelate (HSIC Decoupler) → Regenerate (Field-wise Router)

- **Theory-practice alignment**: Theoretical goals directly map to implementation components

- **High interpretability**: Non-arbitrary, purposeful architecture design

**Theoretical Rigor**

- **Comprehensive proofs**: Each component mathematically justified (A.1: spectral convergence, A.2: rank additivity, A.3: router rank bounds)

- **Provable guarantees**: Positions RoBlock as a provably rank-preserving architecture

**Scalability Validation**

- **Consistent performance**: Maintains superior AUC and Information Abundance (IA) across depth and width scaling on Criteo and Avazu datasets

- **Resists degradation**: Successfully prevents IA decay observed in baseline models

**Appendix Completeness**

- **Thorough documentation**: Full proofs, robustness analysis, computational complexity (A.4.4), and standard deviation results (Tables 5–9)

- **Addresses credibility**: Covers stability, efficiency, and reproducibility concerns

**Weaknesses:**

**HSIC Regularization Sensitivity**

- **Limited exploration**: Only tests β ∈ {0, 0.1, 0.5, 1.0} without comprehensive sensitivity analysis

- **Unclear boundaries**: Performance behavior at extreme β values (→0 or large) unexplored

- **Missing guidance**: No clear recommendations for practitioners on optimal regularization strength selection

**Dataset Diversity and External Validity**

- **Narrow scope**: Limited to two CTR-focused tabular datasets (Criteo and Avazu)

- **Generalizability unknown**: Unclear if anti-collapse mechanisms transfer to:
  - Non-tabular data
  - Multi-modal embeddings (text, vision)
  - Other recommendation domains beyond CTR prediction

**Questions:**

none

---

> ### Author Response · Authors · 2025-11-21
> **Response to Reviewer mbei**
>
> Dear Reviewer mbei,
>
> We thank the reviewers for their valuable time and constructive feedback. Detailed responses to your concerns are provided below, and we would be glad to further engage during the discussion phase if needed.
>
> ---
>
> > (W1) Sensitivity analysis of HSIC regularization with respect to β.
>
> Thank you for your suggestion. We have added the corresponding ablation study in the updated submission (see Appendix A.5.3), where we comprehensively evaluate the impact of β across a **broad parameter range**: {0, 0.1, 0.5, 1, 10, 100}. The results reveal the following insights:
> - With **β = 0** (i.e., **without HSIC regularization**), the model exhibits clear performance degradation, with a performance drop of **0.0008–0.001** observed in RoBlock across both datasets and multiple scale-up factors, confirming the **effectiveness of incorporating HSIC regularization**.
> - The best performance is typically achieved with **β = 1** (and occasionally with 0.5 or 10), while extremely small (0.1) or large (100) values result in minor drops of **0.0001–0.0004**. This indicates that **RoBlock is not overly sensitive to β, and a simple setting of β = 1 provides a robust and effective choice in practice**.
>
> ---
>
> > (W2) Dataset Diversity and External Validity.
>
> Thank you for the valuable comment. We agree that extending our framework to more diverse datasets and recommendation domains is a promising direction for future work. However, we would like to emphasize that focusing on CTR precition using CTR-oriented datasets (Criteo and Avazu) **do not diminish the contributions of this work**:
> - **In-depth study over broad coverage:** Our work addresses a **previously underexplored challenge**, namely ***joint the depth- and width-wise embedding collapse in depth-scaled recommenders***. While prior work [1] tackled **width-wise** collapse in shallow models, we show that in deep recommenders, collapse propagates **across both depth and width**, making previous solutions insufficient. This deeper characterization forms the conceptual foundation of our study.
> - **Practical significance:** Both Criteo and Avazu are **widely used, large-scale industrial CTR benchmarks that serve as standard testbeds for evaluating real-world recommender systems** [2]. The Criteo dataset consists of ad click data collected over a week by the global advertising company Criteo, while the Avazu dataset contains approximately 10 days of labeled click-through data on mobile advertisements collected by Avazu Inc. Both datasets contain over 40 million records and hundreds of features, reflecting the complexity of real-world scenarios. Thus, our experiments are not only methodologically sound but also carry strong practical implications for scalable and deployable recommendation models.
>
> ---
>
> **References**
>
> [1] On the embedding collapse when scaling up recommendation models. ICML'24
>
> [2] Bars: Towards open benchmarking for recommender systems. SIGIR'22

---

> ### Author Response · Authors · 2025-11-26
> **Looking forward to your reply**
>
> Dear Reviewer mbei,
>
> We wanted to kindly follow up on our rebuttal. We are grateful for your careful and constructive feedback, which has genuinely helped strengthen our work. We recognize the significant responsibilities reviewers carry during this period and sincerely appreciate your time. Please do not hesitate to let us know if any concerns remain.
>
> Best regards,
>
> Authors

---

### Official Review · Reviewer_MQVk · 2025-10-27

**Soundness:** 3
**Presentation:** 3
**Contribution:** 3
**Rating:** 6
**Confidence:** 4

**Summary:**

This paper addresses the embedding rank collapse issue that occurs when recommender models are scaled in both width and depth.
The authors propose RoBlock, a modular block composed of Rank-1 normalization, HSIC-based embedding decoupling, and a multi-head router, which jointly maintain high-rank embeddings during training.
Theoretical analysis and experiments on Criteo and Avazu show that RoBlock mitigates collapse and improves AUC over xDeepFM, DCNv2, and DHEN.

**Strengths:**

* Clearly identifies a key limitation in deep/wide scaling of recommenders.

* Provides a well-structured and interpretable module design.

* Strong empirical results that match theoretical motivation.

* Theoretical reasoning supports the design choices.

**Weaknesses:**

* Computational cost and ablation results are not thoroughly discussed.

* Released code appears incomplete ("The requested file is not found."), making reproduction difficult.

**Questions:**

How much additional cost does RoBlock introduce compared to other model architectures, such as DCNv2?

The reported DNN baseline on Avazu seems unusually low, was the baseline model properly tuned (e.g., embedding size, learning rate, dropout, optimizer)

---

> ### Author Response · Authors · 2025-11-21
> **Response to Reviewer MQVk**
>
> Dear Reviewer MQVk,
>
> We sincerely appreciate your time and thoughtful feedback. Detailed responses to your concerns are given below, and we remain fully open to further discussion during the discussion period.
>
> ---
>
> > (W1) Computational cost and ablation results are not thoroughly discussed.
>
> > (Q1) How much additional cost does RoBlock introduce compared to other model architectures, such as DCNv2?
>
> We thank the reviewer for the helpful comments regarding efficiency. In the updated submission, we added a visualized efficiency assessment and discussion (Appendix A.5.1). These results compare RoBlock with both shallow baselines (using 1-layer variant) and deeper baselines (across multipe layers) on 10× embedding setting (embedding dimension = 100), where all baselines are multi-embedding variants for fairness.
>
> We observe that RoBlock delivers solid efficiency. (1) For **GPU memory**, the 1-layer RoBlock ranks **4th** lowest on both datasets among shallow baselines, **reducing memory usage by up to 5%** compared with the strongest AUC baseline xDeepFM, while incurring a minimal increase of **less than 4%** compared with another strong baseline, DCNv2. The multi-layer variants rank **2nd lowest** among deep baselines and **reduce memory by up to 6%** compared with the strongest baseline DHEN. (2) For **training time per epoch**, the 1-layer RoBlock ranks **2nd fastest** on Criteo and **3rd** on Avazu, with up to **24% speedup** over DCNv2 on Criteo and up to **13% speedup** over xDeepFM on Avazu. The multi-layer versions achieve the fastest training speed among deep baselines, with up to **20% reduction in time** compared with DHEN. (3) For **inference latency**, because both rank-1 update and HSIC are **training-only** and **removed at inference**, they introduce no inference overhead. As a result, RoBlock delivers strong efficiency, **staying within 2% of the fastest baselines** in both shallow and deep groups. In summary, our RoBlock demonstrates practical efficiency while providing substantial modeling benefits.
>
> ---
>
> > (W2) Released code appears incomplete ("The requested file is not found."), making reproduction difficult.
>
> We sincerely apologize for the inconvenience and would like to clarify that the issue was likely caused by a temporary error on the anonymous-science4open website during the recent intensive submission period. We have confirmed that the error has been resolved, and the code is now accessible. The anonymous code contains RoBlock implementation for review purposes only, and the full codebase will be publicly released upon acceptance.
>
> ---
>
> > (Q2) The reported DNN baseline on Avazu seems unusually low, was the baseline model properly tuned (e.g., embedding size, learning rate, dropout, optimizer)
>
> We believe the observed differences are due to **our experimental settings compared to the FuxiCTR library**. Specifically, (1) we set the base embedding size (i.e., the item "base" in the data tables) to 10 for **all models** on Avazu, which is smaller than the value of 16 used in FuxiCTR, (2) we finetune only **model-specific** hyperparameters (e.g., number of layers in the MLP_block, activation functions, and hidden units) based on the FuxiCTR-tuned models, while **universal components** such as learning rate, regularization, dropout, and optimizer are **kept uniform across all models for fairness**, and (3) for better statistical reliability, we run each experiment with 20 different random seeds and report the averaged test results.
>
> **Under this controlled evaluation protocol aimed at ensuring fairness across models, performance fluctuations across baselines are natural and should be expected**. We observe that some baselines (e.g., CIN, CrossNet) report higher results than those in FuxiCTR on one or both datasets, some (e.g., xDeepFM) perform higher on one dataset but lower on another, and some (e.g., DNN, NFM) yield lower results on both. We believe that such variations do not impact the overall significance of our findings and contributions.

---

> > ### Comment · Reviewer_MQVk · 2025-11-27
> >
> > Thank you for the detailed rebuttal. My main concerns are largely addressed, and I will keep the positive score I have already assigned.
> > I strongly encourage the authors to include the discussion of computational cost directly in the main paper rather than only in the appendix, as this aspect is important for assessing the practical utility of the proposed method.

---

> > > ### Author Response · Authors · 2025-11-27
> > > **Thank You and Manuscript Update**
> > >
> > > Dear Reviewer MQVk,
> > >
> > > **Thank you very much for your positive feedback and rating on our submission and rebuttal**. We appreciate your suggestions and have updated our manuscript to include key discussion regarding computational cost in the main text, while adhering to the 10-page limit for the ICLR rebuttal phase. We believe that these updates further highlight the significance of our research.
> > >
> > > Best regards,
> > >
> > > Authors

---

> ### Author Response · Authors · 2025-11-26
> **Looking forward to your reply**
>
> Dear Reviewer MQVk,
>
> We wanted to kindly follow up on our rebuttal. We are grateful for your careful and constructive feedback, which has genuinely helped strengthen our work. We recognize the significant responsibilities reviewers carry during this period and sincerely appreciate your time. Please do not hesitate to let us know if any concerns remain.
>
> Best regards,
>
> Authors

---

### Official Review · Reviewer_9TjC · 2025-10-27

**Soundness:** 3
**Presentation:** 3
**Contribution:** 2
**Rating:** 4
**Confidence:** 4

**Summary:**

This work addresses the problem of embedding rank collapse in large-scale recommendation systems where the embeddings lose expressive power as the model scales (both in depth and width). To solve this issue, the authors propose RoBlock, a modular design which helps maintain the dimensionality of the original embedding space and mitigate collapse. RoBlock adopts three core designs: (a) rank-1 update normalization to restore the singular values of the embeddings, (b) decoupled embeddings to encode different properties of the data, and (c) HSIC regularization to encourage diversity across the decoupled embeddings. Each of these has associated theoretical analysis on how they help preserve the rank of the embeddings. Empirically, the method is able to show robustness to different depths and widths, with some gains in performance.

**Strengths:**

I will start with the strengths of the work.

1. I think the work studies an important problem on embedding collapse with clear motivation as to why we should care about it.
2. The authors offer a series of modular designs within RoBlock which can be easily adapted to different settings. I also appreciate that each is motivated with theoretical analysis.
3. The authors offer strong empirical evidence that they can remedy the embedding collapse problem (which is different than improving performance), particularly as the models scale across width or depth dimensions.

**Weaknesses:**

Despite the strengths, there are quite a few areas that lack justification or deeper analysis.

1. Comparison to Previous Collapse Mitigation: The authors cite a variety of methods that study embedding collapse. Moreover, there are works in the space of recommenders that also work on this problem. Despite this, the authors do not offer any direct empirical analysis to them. Moreover, many of them tend to be much simpler compared to RoBlock, containing one component/loss, thus it is important to demonstrate that RoBlock warrants its multiple designs by out-performing them.
2. Theory: While the authors provide theory to support the maintaining of rank, a clear gap is connecting rank to performance. While the authors intuitively argue that higher rank is better, it is not obvious to me that collapse is always problematic. It would be good for the authors to address this disconnect and make a rigorous argument connecting rank to performance.
3. Runtime Analysis: No empirical runtime analysis is provided for the work. It would be great to have a sense of how much computational overhead the method requires as compared to the previous methods.
4. Performance: This is more minor, but it can be hard to assess the significance of the performance improvements in Table 1. When reading the original paper that proposed Criteo and Avazu, it seems as though these numbers are comparable to what they original achieve. Can the authors further justify why improvements at the 3rd/4th decimal point are meaningful?

**Questions:**

1. Can the authors provide a deeper analysis to methods that mitigate embedding collapse? As of now, the baselines are relatively simple.
2. Can the authors better connect matrix rank to performance?
3. Can the authors provide an empirical runtime analysis compared to the other methods?
4. Can the authors justify the significance of the performance? I will acknowledge that I see the standard deviations (I would like to see those in the main table, it is unclear why they are int he appendix), but that doesn't change the fact that the performance appears marginal.

---

> ### Author Response · Authors · 2025-11-21
> **Response to Reviewer 9TjC (Part 1)**
>
> Dear Reviewer 9TjC,
>
> We greatly appreciate your careful review and insightful comments. Our detailed responses to your concerns are provided below, and we are happy to engage with any additional questions during the discussion phase.
>
> ---
>
> > (W1) Comparison to Previous Collapse Mitigation: The authors cite a variety of methods that study embedding collapse. Moreover, there are works in the space of recommenders that also work on this problem. Despite this, the authors do not offer any direct empirical analysis to them. Moreover, many of them tend to be much simpler compared to RoBlock, containing one component/loss, thus it is important to demonstrate that RoBlock warrants its multiple designs by out-performing them.
>
> > (Q1) Can the authors provide a deeper analysis to methods that mitigate embedding collapse? As of now, the baselines are relatively simple.
>
> ***Comparison with Latest Works.*** Thank you for your comments. We provide additional experiments comparing RoBlock with three recent methods proposed to address embedding collapse in recommenders [1,2,3], evaluated under the largest 10× embedding size. These methods introduce either **new component** (GE4Rec [1]) or **new loss** (CorrLoss [2], CovLoss [3]) to **conventional recommenders**. Following [1,4], we adopt DCNv2 and xDeepFM as backbones for these methods and apply multi-embedding. To ensure fairness, we control parameter size of RoBlock and all baselines within 500M ± 1%. RoBlock is set 1 layer and scaled only by increasing components m. As shown below, RoBlock delivers **clearly better AUC** than all counterparts, up to **+0.00027** on Criteo and **+0.001** on Avazu over the strongest competitor, GE4Rec, while also **reducing training time by 15%–20%** relative to it.
>
> | | #Params | AUC on Criteo | Time/Epoch on Criteo | AUC on Avazu | Time/Epoch on Avazu |
> |:---:|:---:|:---:|:---:|:---:|:---:|
> | CovLoss + DCNv2 | 498.6M | 0.81427(1e-4) | 17.08min | 0.79577(2e-4) | 14.37min |
> | CovLoss + xDeepFM | 504.3M | 0.81444(2e-4) | 20.13min | 0.79622(2e-4) | 15.91min |
> | CorrLoss + DCNv2 | 498.6M | 0.81451(2e-4) | 17.19min | 0.79618(3e-4) | 14.71min |
> | CorrLoss + xDeepFM | 504.3M | 0.81432(2e-4) | 19.53min | 0.79601(2e-4) | 16.59min |
> | GE4Rec + DCNv2 | 500.1M | 0.81536(3e-4) | 24.34min | 0.79644(2e-4) | 21.61min |
> | GE4Rec + xDeepFM | 498.8M | 0.81513(3e-4) | 26.29min | 0.79665(3e-4) | 20.32min |
> | RoBlock | 499.4M | **0.81561**(2e-4) | 19.99min | **0.79764**(3e-4) | 17.13min |
>
> ***On Addressing Depth-wise Collapse.*** We also test whether CorrLoss and CovLoss can handle **depth-wise collapse** by applying them on DHEN using 5 layers and 10× size (**GE4Rec is incompatible**). Results below show that while the losses slightly increase IA and AUC, their improvements are **small and far behind RoBlock**.
> Moreover, we find that **RoBlock is orthogonal and easily composable with these methods**: adding CorrLoss on top of RoBlock directly delivers an additional **+0.00043 (on Criteo)** and **+0.00061 (on Avazu)** in AUC.
>
> |  | IA on Criteo | AUC on Criteo | IA on Avazu | AUC on Avazu |
> |---|---|---|---|---|
> | DHEN (origin) | [30.7, 22.6, 15.6, 13.5, 10.3] | 0.81435(3e-4) | [21.6, 17.2, 13.0, 10.8, 9.3] | 0.79479(2e-4) |
> | DHEN + CorrLoss | [32.3, 23.7, 17.3, 15.1, 13.1] | 0.81450(3e-4) | [22.2, 18.4, 15.0, 12.7, 10.3] | 0.79513(3e-4) |
> | DHEN + CovLoss | [31.7, 24.2, 16.8, 14.4, 12.5] | 0.81439(2e-4) | [21.9, 17.9, 14.3, 12.1, 9.9] | 0.79536(2e-4) |
> | RoBlock | **[37.7, 36.3, 33.8, 31.8, 28.9]** | **0.81480**(2e-4) | **[23.1, 22.5, 21.1, 19.7, 18.3]** | **0.79702**(1e-4) |
>
> ***Analysis and Summary.*** These results demonstrate that RoBlock’s design is well justified, **outperforming simpler loss- or component-based methods both in effectiveness and efficiency**. More importantly, on **deep recommenders** where joint depth- and width-wise collapse occurs, these recent methods **still suffer from severe depth-wise collapse**, whereas RoBlock effectively mitigates this issue and delivers superior performance gains. Overall, **these results serve as complementary evidence that underscores the necessity of RoBlock’s targeted design for this underexplored problem, while leaving intact the self-contained structure of the main paper that already establishes this key focus**.
>
> ---
>
> **References**
>
> [1] From feature interaction to feature generation: a generative paradigm of CTR prediction models. ICML'25
>
> [2] Enhancing CTR prediction with de-correlated expert networks.
>
> [3] Crocodile: cross experts covariance for disentangled learning in multi-domain recommendation. CIKM'25
>
> [4] On the embedding collapse when scaling up recommendation models. ICML'24

---

> ### Author Response · Authors · 2025-11-21
> **Response to Reviewer 9TjC (Part 2)**
>
> > (W2) Theory: While the authors provide theory to support the maintaining of rank, a clear gap is connecting rank to performance. While the authors intuitively argue that higher rank is better, it is not obvious to me that collapse is always problematic. It would be good for the authors to address this disconnect and make a rigorous argument connecting rank to performance.
>
> > (Q2) Can the authors better connect matrix rank to performance?,
>
> Thanks for the suggestion. We clarify the connection here: our main conclusion is that, *for **trained** recommenders, preserving embedding rank (spectrum) leads to better performance*.
>
> - First, our discussion applies only to **trained** models. To clarify this, we compare DHEN and Wukong under random initialization versus after training, using a 5x size. **Random matrices are typically full-rank** [2], so untrained models naturally appear high IA. However, **their AUC is extremely low** as the representations have not learned any task-relevant knowledge. In contrast, trained models show lower IA but much higher AUC. This shows that the spectrum before training is not meaningful, and our analysis concerns trained models.
>
> |  | IA on Criteo | AUC on Criteo | IA on Avazu | AUC on Avazu |
> |:---:|:---:|:---:|:---:|:---:|
> | DHEN (5x, w/o training) | [35.3, 28.3, 25.8, 20.8, 14.9] | 0.43125 | [22.1, 19.8, 17.7, 15.0, 12.4] | 0.41145 |
> | DHEN (5x, w/ training) | [30.9, 22.3, 14.2, 11.2, 9.7] | 0.81406 | [21.8, 17.9, 13.8, 10.2, 8.5] | 0.79456 |
> | Wukong (5x, w/o training) | [34.2, 30.5, 24.5, 19.6, 15.5] | 0.36722 | [21.6, 18.4, 16.1, 14.2, 11.7] | 0.45114 |
> | Wukong (5x, w/ training) | [29.1, 17.9, 13.2, 9.1, 7.9] | 0.81380 | [20.7, 16.2, 11.1, 9.5, 7.8] | 0.79418 |
>
> - Second, for the **trained** models, preserving the spectrum is beneficial. As reported in Sec. 5.3, **models with less collapse consistently achieve better AUC**. Moreover, according to [1], once collapse appears in the embeddings, the interaction modules amplify it, driving the embeddings toward a low-rank dilemma. This limits the effective dimensionality of the representations and reduces model capacity [3,4]. Motivated by these insights, our RoBlock aims to keep the spectrum balanced and prevent collapse during training, preserving informative embeddings that can be fully exploited by the interaction module, ultimately leading to better performance.
>
> In summary, although a theory–practice gap exists, our experiments clearly show that **preserving rank (spectrum) is tightly coupled with improved performance in trained models**, reinforcing the core focus and novelty of this work.
>
> ---
>
> > (W3) Runtime Analysis: No empirical runtime analysis is provided for the work. It would be great to have a sense of how much computational overhead the method requires as compared to the previous methods.
>
> > (Q3) Can the authors provide an empirical runtime analysis compared to the other methods?
>
> We thank the reviewer for the helpful comments regarding efficiency. In the updated submission, we added a visualized efficiency assessment and discussion (Appendix A.5.1). These results compare RoBlock with both shallow baselines (using 1-layer variant) and deeper baselines (across multipe layers) on 10× embedding setting (embedding dimension = 100), where all baselines are multi-embedding variants for fairness.
>
> We observe that RoBlock delivers solid efficiency. (1) For **GPU memory**, the 1-layer RoBlock ranks **4th lowest** on both datasets among shallow baselines, **reducing memory usage by up to 5%** compared with the strongest AUC baseline xDeepFM, while incurring a minimal increase of **less than 4%** compared with another strong baseline, DCNv2. The multi-layer variants rank **2nd lowest** among deep baselines and **reduce memory by up to 6%** compared with the strongest baseline DHEN. (2) For **training time per epoch**, the 1-layer RoBlock ranks **2nd fastest** on Criteo and **3rd** on Avazu, with up to **24% speedup** over DCNv2 on Criteo and up to **13% speedup** over xDeepFM on Avazu. The multi-layer versions achieve the fastest training speed among deep baselines, with up to **20% reduction in time** compared with DHEN. (3) For **inference latency**, since both **rank-1 update and HSIC are training-only and removed at inference, they introduce no inference overhead**. As a result, RoBlock delivers strong efficiency, **staying within 2% of the fastest baselines** in both shallow and deep groups. In summary, RoBlock demonstrates practical efficiency while providing substantial modeling benefits.
>
> ---
>
> **References**
>
> [1] On the embedding collapse when scaling up recommendation models. ICML'24
>
> [2] High-dimensional probability.
>
> [3] Understanding dimensional collapse in contrastive self-supervised learning. ICLR'22
>
> [4] Towards understanding and mitigating dimensional collapse in heterogeneous federated learning. ICLR'23

---

> ### Author Response · Authors · 2025-11-21
> **Response to Reviewer 9TjC (Part 3)**
>
> > (W4) Performance: This is more minor, but it can be hard to assess the significance of the performance improvements in Table 1. When reading the original paper that proposed Criteo and Avazu, it seems as though these numbers are comparable to what they original achieve. Can the authors further justify why improvements at the 3rd/4th decimal point are meaningful?
>
> > (Q4) Can the authors justify the significance of the performance? I will acknowledge that I see the standard deviations (I would like to see those in the main table, it is unclear why they are int he appendix), but that doesn't change the fact that the performance appears marginal.
>
> Thank you for the comments. Due to ICLR’s page limit, some results appear in the appendix and will be moved back to the main text upon acceptance.
>
> Next, we would like to clarify that, in the context of CTR prediction, **improvements at the 3rd or 4th decimal place are indeed meaningful and significant**:
> - First, as highlighted by a reputable study [1] (see Sec. 6.6), ***"0.001 absolute AUC gain is significant and worthy of model deployment empirically."*** This is consistent with numerous recent mainstream works in CTR prediction (e.g., [1,2,3,4,5,6,7,8,9,10,11]), where reported improvements on standard datasets **typically appear at the 3rd or 4th decimal**. Such fine-grained gains have become the accepted standard for demonstrating meaningful progress in CTR research, and the community widely recognizes that they correspond to real and substantial enhancements in prediction performance.
> - Second, the **scale and complexity** of these datasets are substantially higher than many commonly used public datasets. For example, datasets such as *Frappe* (29K instances, 5K features, 10 fields) or *Movielens_Latest* (2M instances, 90K features, 3 fields) are orders of magnitude smaller. In contrast, **Criteo and Avazu contain 40M–45M instances, 5M–8M features, and 25–40 fields**, making them far more challenging and reflective of real production-scale CTR scenarios. Under such complexity, even small improvements translate into large user-level or revenue-level gains in real industrial systems.
>
> Therefore, the improvements achieved by RoBlock, mostly in the 3rd decimal points and occasionally the 4th, are considered both meaningful and valuable, and they are consistent with what is expected and accepted in the CTR prediction literature.
>
> ---
>
> **References**
>
> [1] Deep interest network for click-through rate prediction. KDD'18
>
> [2] DLF: enhancing explicit-implicit interaction via dynamic low-order-aware fusion for ctr prediction. SIGIR'25
>
> [3] Towards unifying feature interaction models for click-through rate prediction. ECMLPKDD'25
>
> [4] ORIC: feature interaction detection through online random interaction chains for click-through rate prediction. TKDD'25
>
> [5] Fusion matters: learning fusion in deep click-through rate prediction models. WSDM'25
>
> [6] A collaborative ensemble framework for ctr prediction.
>
> [7] DELTA: dynamic embedding learning with truncated conscious attention for ctr prediction.
>
> [8] Enhancing ctr prediction with de-correlated expert network.
>
> [9] From feature interaction to feature generation: a generative paradigm of ctr prediction models. ICML'25
>
> [10] On the embedding collapse when scaling up recommendation models. ICML'24
>
> [11] FCN: fusing exponential and linear cross network for click-through rate prediction.

---

> > ### Comment · Reviewer_9TjC · 2025-11-25
> > **Thank you!**
> >
> > I want to thank the authors for their thorough rebuttal. I have gone through the discussion and appreciate their effort to address my questions and critiques. Based on the added citations, I do believe the results are significant, especially with the added experiments.
> >
> > I think it would be worth thinking a bit more regarding the high IA, even for trained recommenders. I do agree with the fact that random matrices will have high IA without being informative, but there is also a ton of work in recommender systems that demonstrate the importance of hashing/collisions as a means to improve performance. It would seem rank collapse is an implicit way to encourage collisions, especially for extremely large datasets. Discussing these caveats with high IA, in my opinion, would strengthen the paper.
> >
> > Overall, I will raise my score to reflect these additions.

---

> > > ### Author Response · Authors · 2025-11-26
> > > **Appreciation and Preliminary Exploration**
> > >
> > > **We sincerely thank you for the thoughtful feedback and for raising our score**. This encouragement motivates us to explore various aspects in future work, such as the *hashing collision* you mentioned. As a **preliminary check**, we conducted a quick test on DHEN (5-layer, 5× embedding size), comparing its *multi-embedding* (ME) [1] variant (the baseline in our paper) with a version that applies *Collision-Weighted Lookups* (CWL) [2], a recent technique designed to address hash collision.
> > >
> > > Specifically, applying CWL yields comparable IA and AUC compared with ME. **Both ME and CWL aim to enhance embedding capacity in the initial embedding layer** but differ in approach: ME parallelizes multiple independent embedding tables to mitigate collapse, while CWL introduces learnable adjustments within a single table to reduce collision. **However, neither method can address the depth-wise collapse that emerges in deeper layers**. Existing proposals focus mainly on shallow models—such as the representative DCNv2 used in [1] and [2]—and thus only refine low-rank collapse or hashing collision at the initial embedding layer. In contrast, **the underexplored depth-wise collapse in each layer of deep recommenders arises from feature interactions and complex transformations [1], making initial-layer refinements insufficient**. Overall, the quick experiments imply that collapse relates to collision in the initial embeddings, but **the depth-wise collapse likely requires separate solutions—such as the rank (spectrum) preservation used in our RoBlock**.
> > >
> > > |  | IA on Criteo | AUC on Criteo | IA on Avazu | AUC on Avazu |
> > > |:---:|:---:|:---:|:---:|:---:|
> > > | DHEN + CWL [2] | [29.8, 21.5, 13.2, 10.8, 9.0] | 0.81398(3e-4) | [21.2, 17.4, 13.9, 11.2, 9.3] | 0.79463(3e-4) |
> > > | DHEN + ME [1] (baseline in paper) | [30.9, 22.3, 14.2, 11.2, 9.7] | 0.81406(2e-4) | [21.8, 17.9, 13.8, 10.2, 8.5] | 0.79456(3e-4) |
> > >
> > > **In future work, we will continue exploring scaling mechanisms for recommenders, including aspects such as hash collisions.**
> > >
> > > ---
> > > **References:**
> > >
> > > [1] On the embedding collapse when scaling up recommendation models. ICML'24
> > >
> > > [2] DCN²: interplay of implicit collision weights and explicit cross layers for large-scale recommendation. AdKDD in KDD'25

---

> > > > ### Comment · Reviewer_9TjC · 2025-11-26
> > > > **Thanks!**
> > > >
> > > > Thank you for the additional follow up even after I raised my score. These results are interesting and would be great to explore further in follow up work. I have no further comments or questions!

---

### Official Review · Reviewer_z8Nc · 2025-11-01

**Soundness:** 3
**Presentation:** 3
**Contribution:** 2
**Rating:** 4
**Confidence:** 4

**Summary:**

The idea of RoBlock is quite clear — it treats the common embedding rank collapse problem in recommender systems as the main challenge to solve.
The authors argue that, after multiple layers of feature interactions, the embedding information tends to concentrate in a few dominant dimensions, leading to reduced rank and degraded representational capacity.
To address this, they design a modular structure where each layer is a RoBlock module that performs several operations:
first, Rank-1 Update Normalization to rebalance the singular values and prevent energy concentration;
second, HSIC regularization to encourage independence among multiple sub-embeddings and improve rank;
third, a field-wise router to dynamically combine features from different fields;
and finally, residual connections to stabilize information flow between layers.
Overall, the motivation is sound and the system design is consistent.
However, the novelty is somewhat limited — similar ideas such as spectral balancing, HSIC-based independence, router gating, and residual fusion have appeared in prior works.
RoBlock’s main contribution lies in integrating these mechanisms into a unified modular framework, which is well-structured and has some engineering value.

**Strengths:**

Clear problem definition and motivation.
The paper focuses on the embedding rank collapse issue in recommender systems, which is indeed an important challenge when scaling deep recommendation models.

Well-structured overall design.
By combining Rank-1 Update Normalization, HSIC-based decoupling, field-wise routing, and residual connections into a modular framework, the method provides a systematic structure for mitigating collapse.

Consistent theoretical and empirical support.
Theoretical analysis on spectral balancing and independence, together with the IA visualization results, reasonably supports the proposed design.

**Weaknesses:**

Limited novelty.
Most components of the proposed method have already appeared in prior works; the contribution mainly lies in integrating existing ideas rather than introducing a new algorithmic principle.

No discussion of efficiency.
The model involves computationally heavy operations (e.g., rank-1 update, HSIC regularization, dynamic routing), yet the paper does not evaluate or discuss their efficiency.

Unclear practical value.
Although the design may theoretically alleviate embedding degradation, the high computational cost raises concerns about its feasibility in real-world recommender systems.

**Questions:**

Could the authors provide an evaluation of the computational efficiency of RoBlock?
In particular, it would be helpful to report training time, inference latency, and memory usage compared to baselines such as xDeepFM or DHEN.
Since the proposed modules (Rank-1 Update, HSIC regularization, and field-wise router) are computationally intensive, understanding their actual runtime overhead is important for assessing the model’s practicality.

Have you considered using approximations or lightweight variants (e.g., simplified normalization or reduced HSIC sampling) to balance effectiveness and efficiency?

How does the model scale when applied to larger datasets or higher embedding dimensions?

---

> ### Author Response · Authors · 2025-11-21
> **Response to Reviewer z8Nc (Part 1)**
>
> Dear Reviewer z8Nc,
>
> We truly appreciate your time and insights. Below, we have addressed the concerns you highlighted in detail, and we are happy to continue the conversation should additional points require clarification.
>
> ---
>
> > (W1) Limited novelty.
>
> We hope to clarify that our novelty lies in **identifying and addressing an underexplored research problem**—namely, the ***joint the depth- and width-wise embedding collapse in depth-scaled recommenders***. Prior work [1] has shown that shallow recommenders suffer from **width-wise** collapse at the initial embedding and proposed multi-embedding as a remedy. However, modern deep recommenders are built with a block-by-block (layered) architecture, and we show for the first time that embedding collapse propagates across **both width and depth**, making multi-embedding alone insufficient. This new characterization underpins our work.
>
> RoBlock is **designed to address the underexplored problem, with each component serving a purposeful role**. First, to **refine the collapse of input embeddings at each depth**, we apply **rank-1 update normalization**, which is originally used for low-rank approximation, preserving a balanced spectrum. Second, inspired by the success of multi-embedding in shallow recommenders, we propose **embedding decoupler regularized by HSIC**, enabling a ***deep-model analog* of multi-embedding** with informative embedding components. Third, guided by interaction collapse theory [1], we apply **embedding regeneration via a field-wise router** to adaptively produce expressive embeddings for interaction modules, ensuring that crucial **feature interactions receive non-collapsed inputs**. Overall, **each design choice directly targets the collapse challenge**, and our experiments validate both effectiveness of these components and strength of RoBlock as a whole.
>
> ---
>
> > (W2) No discussion of efficiency.
>
> > (W3) Unclear practical value.
>
> > (Q1) Could the authors provide an evaluation of the computational efficiency of RoBlock? In particular, it would be helpful to report training time, inference latency, and memory usage compared to baselines such as xDeepFM or DHEN. Since the proposed modules (Rank-1 Update, HSIC regularization, and field-wise router) are computationally intensive, understanding their actual runtime overhead is important for assessing the model’s practicality.
>
> We thank the reviewer for the comments and address the efficiency concerns as follows.
>
> **Supplementary Efficiency Evaluation.** In the updated submission, we have added an extensive efficiency study (Appendix A.5.1). The evaluation compares RoBlock with both shallow baselines (using 1-layer RoBlock) and deep baselines (across multipe layers) on the largest 10× size (embedding dimension = 100), where all baselines are multi-embedding variants for fairness. Overall, RoBlock demonstrates solid and competitive efficiency: (1) For **GPU memory**, the 1-layer RoBlock ranks **4th lowest** on both datasets among shallow baselines, **reducing memory usage by up to 5%** compared with the strongest AUC baseline xDeepFM. The multi-layer variants rank **2nd lowest** among deep baselines and **reduce memory by up to 6%** compared with the strongest baseline DHEN. (2) For **training time per epoch**, the 1-layer RoBlock ranks **2nd fastest** on Criteo and **3rd** on Avazu, with up to **24% speedup** over DCNv2 on Criteo and up to **13% speedup** over xDeepFM on Avazu. The multi-layer versions achieve the **fastest** training speed among deep baselines, with up to **20% reduction in time** compared with DHEN. (3) For **inference latency**, RoBlock shows leading performance overall, **staying within 2% of the fastest baselines** in both shallow and deep groups.
>
> **Analysis on Efficiency Results.** The results show that operations you raised concerns about **do not lead to significant overhead**, consistent with the complexity analysis in Appendix A.4.4 and A.5.2. RoBlock maintains practical efficiency while providing substantial modeling benefits. In particular, two key components, i.e., **rank-1 update normalization and HSIC loss, are used only during training. They are removed at inference, hence incuring no impact on the online computation**. As a result, RoBlock maintains inference latency on par with the fastest baselines. The remaining part of RoBlock, including field-wise router, uses only lightweight MLPs and basic matrix operations, which are easy to deploy in real serving systems. Overall, RoBlock provides practical efficiency rather than incurring heavy computational cost.
>
> ---
>
> **References:**
>
> [1] On the embedding collapse when scaling up recommendation models. ICML'24

---

> ### Author Response · Authors · 2025-11-21
> **Response to Reviewer z8Nc (Part 2)**
>
> > (Q2) Have you considered using approximations or lightweight variants (e.g., simplified normalization or reduced HSIC sampling) to balance effectiveness and efficiency?
>
> Thank you for your comments. According to your suggestion, we experimented with a **reduced HSIC sampling strategy**, where only **a portion of each batch (10% and 50%) is randomly selected for HSIC computation**. As shown below, while this strategy slightly improves efficiency, it leads to a **noticeable drop in both AUC and IA**, and **increased AUC variance** indicates unstable training under extreme reduction. This is because computing HSIC on only subset gives noisy and biased estimate of dependence, leading to unstable and poor training. Moreover, the **little efficiency gain** between the full HSIC ("origin") and 10% sampling suggests that the cost of vanilla HSIC is already small. Hence, **using the original HSIC is both efficient and effective**.
>
> |  | IA on Criteo | AUC(std) on Criteo | (Memory,Time) on Criteo | IA on Avazu | AUC(std) on Avazu | (Memory,Time) on Avazu |
> |:---:|:---:|:---:|:---:|:---:|:---:|:---:|
> | origin | [37.7, 36.3, 33.8, 31.8, 28.9] | **0.81480**(2e-4) | (9.26, 15.02) | [23.1, 22.5, 21.1, 19.7, 18.3] | **0.79702**(1e-4) | (6.56, 14.13) |
> | 50% HSIC | [37.1, 31.3, 29.0, 26.5, 22.2] | 0.81461(3e-4) | (9.05, 14.95) | [22.8, 19.3, 17.7, 15.8, 13.3] | 0.79652(4e-4) | (6.40, 14.07) |
> | 10%HSIC | [34.5, 27.3, 25.2, 18.4, 13.0] | 0.81422(6e-4) | (8.90, 14.89) | [21.3, 18.3, 15.1, 13.5, 11.0] | 0.79625(6e-4) | (6.27, 14.03) |
>
> On the other hand, the rank-1 update normalization, a form of *power method*, **is already a lightweight approximation algorithm**, commonly used to estimate the spectral norm. **This approach has been successfully deployed on large-scale datasets such as *ImageNet1K*** [7], demonstrating that the computational cost is practical and well-suited for real-world settings.
>
> ---
>
> > (Q3) How does the model scale when applied to larger datasets or higher embedding dimensions?
>
> Thank you for your comments. Regarding ***embedding dimensions***, we tested an extreme setting, increasing from **10× (100) to 50× (500)** on a 5-layer RoBlock. This yielded AUC improvements of **0.000477** for Criteo and **0.001102** for Avazu, with a **5%–7%** increase in memory and time, which are **smaller gains but higher relative cost than scaling from 2× (20) to 10× (100)**. These results show that **benefits diminish** once embedding dimension becomes extremely larger than typical recommender settings (which is usually 10 or 16 [1,2,3,4,5,6]), and additional issues such as overfitting may arise. Nevertheless, this does **NOT** affect the key contribution of our work, which is demonstrating **the benefits of mitigating the underexplored joint depth- and width-wise embedding collapse**, as discussed in response to W1. Our experiments cover a wide range of embedding dimensions, from the standard 10 used in mainstream research [1,2,3,4,5,6] to an extraordinary 10× base size (100), which is already far larger than typical settings in the literature. Combined with both performance and efficiency results, this evaluation confirms that RoBlock shows substantial scalability and maintains superior effectiveness.
>
> Regarding ***datasets***, both Criteo and Avazu are widely recognized, large-scale industrial benchmarks, which allow for a comprehensive assessment of RoBlock’s scalability. The results convincingly demonstrate the practical usability of our method.
>
> ---
>
> **References:**
>
> [1] On the embedding collapse when scaling up recommendation models. ICML'24
>
> [2] Towards unifying feature interaction models for click-through rate prediction. ECMLPKDD'25
>
> [3] MATT-CTR: unleashing a model-agnostic test-time paradigm for CTR Prediction with confidence-guided inference paths.
>
> [4] Bars: Towards open benchmarking for recommender systems. SIGIR'22
>
> [5] Topic guided multi-faceted semantic disentanglement for CTR prediction. MM'25
>
> [6] From feature interaction to feature generation: a generative paradigm of CTR prediction models. ICML'25
>
> [7] Preventing dimensional collapse in self-supervised learning via orthogonality regularization. NIPS'24

---

> ### Author Response · Authors · 2025-11-26
> **Looking forward to your reply**
>
> Dear Reviewer z8Nc,
>
> As the discussion period will soon end and it has been over five days since our rebuttal, we would kindly like to ask whether our rebuttal has addressed your concerns. We have carefully responded to points regarding our preliminary study, analysis methods, and other design details.
>
> We would greatly appreciate your feedback on whether our responses meet your expectations. If any questions remain, we are eager to continue the discussion. If our rebuttal has sufficiently addressed your feedback, we hope you will consider re-evaluating our paper.
>
> Thank you again for your valuable time and thoughtful review. We look forward to your response.
>
> Best regards,
>
> Authors

---

### Author Response · Authors · 2025-12-01
**Rebuttal Summary: Addressing of Reviewer Concerns, Contributions, and Strengths**

Dear PCs, SACs, ACs, and Reviewers,

We sincerely thank you for the time and effort in reviewing and discussing our submission. Below, we summarize the outcomes of the review and discussion process.

---

## **1. Addressing of Reviewer Concerns**

We provided detailed responses and targeted revisions to address all reviewer concerns. **Two** reviewers engaged in the discussion, and **both confirmed their concerns were resolved**, leading to a **score increase** to (**6**,**6**,**6**,**4**). We recap the discussion outcomes below:

|Reviewer|Score|Response|Discussion Outcomes|
|:-:|:-:|:-:|:-|
|Reviewer z8Nc|4|**No Response**|The reviewer provided **no response** during the discussion. **Their concerns largely mirror those raised by other reviewers**, and were addressed as below:|
||||**1. Novelty:** We clarified that our core novelty lies in identifying the **underexplored joint depth- and width-wise embedding collapse problem**, and developing **problem-driven, theoretically grounded modules** to the newly identified issue.|
||||**2. Practical Value and Efficiency:** Same concern was also raised by Reviewers 9TjC and MQVk, and was addressed by the **additional efficiency study** in the revised manuscript (main text + Appendix A.5.1). Both Reviewers 9TjC and MQVk **praised this update**.|
||||**3. Module Complexity and Experiment Scale:** Similar concern was raised by Reviewer 9TjC and was addressed by empirical results and formal analyses, which show **low cost** of RoBlock’s modules and **high scalability** of RoBlock. We emphasized that **our experiments run at very large scales**, using industry-scale benchmarks and embedding sizes from typical prior sizes to 10x larger.|
|Reviewer 9TjC|4 → **6**|[**Score Upgraded**] 17:41, 25 Nov (UTC); [**No Further Concerns**] 19:36, 26 Nov (UTC)|The reviewer **upgraded the rating to positive (6)** and indicated their concerns were addressed.|
|Reviewer MQVk|6 → **6**|[**Concerns Addressed**] 13:24, 27 Nov (UTC)|The reviewer **confirmed their positive score (6)** and indicated their concerns were addressed. We further revised the manuscript based on the follow-up suggestion.|
|Reviewer mbei|6|**No Response**|The reviewer provided **no response** during the discussion. Their concerns were resolved as below:|
||||**1. Parameter Analysis on β:** We added analyses to the manuscript (Appendix A.5.3). These additions provide intuitive **guidance on choosing β** and highlight **parameter robustness** of RoBlock.|
||||**2. Research Scope:** We clarified that CTR prediction is a **practical and important real-world task**, and benchmarks we use are **widely adopted, industry-scale datasets for realistic evaluation**.|

---

## **2. Summary of Paper Contributions**

We restate below the contributions that reviewers consistently recognized as **novel and valuable**:
- We identify an **underexplored joint depth-wise and width-wise embedding collapse problem** in deep recommenders, extending the recently examined **width-wise collapse** in **traditional** recommenders [1,2,3] to **modern deep architectures**.
- We propose **RoBlock as a principled remedy for the identified problem**. Each module of RoBlock is **carefully crafted with clear motivation**, maintains **low cost for practical deployment**, and is **supported by consistent theoretical justification**.
- We conduct **extensive and targeted experiments** on **industry-scale benchmarks**, ***Criteo*** and ***Avazu***, to show the strong capability, scalability, and efficiency of RoBlock. RoBlock delivers superior performance, achieving **up to 0.002 AUC gains**—a substantial improvement in CTR prediction [4]—over strong baselines, together with **up to 5% memory savings and 24% faster training**. These results underscore **the necessity of addressing the newly identified collapse** and **pave the way for further advances in deep recommenders**.

---

## **3. Recognized Strengths across ALL Reviewers**

The **strengths** listed below were **unanimously recognized by ALL reviewers**, demonstrating both rigor and exceptional contribution of our work.
- A clear, novel, and important research problem with well-structured motivation.
- A well-structured, interpretable model design with purposeful components.
- Strong theoretical foundations and alignment between problem formulation and proposed solution.
- Extensive and consistent empirical validations.

---

**We believe these updates and clarifications further strengthen the paper and highlight its significance for both research and real-world applications.**

Best regards,

Authors

---
**References:**

[1] *On the embedding collapse when scaling up recommendation models*. In ICML, 2024.

[2] *Understanding embedding scaling in collaborative filtering*. TMLR (under review).

[3] *From feature interaction to feature generation: a generative paradigm of ctr prediction models*. In ICML, 2025.

[4] *Deep interest network for click-through rate prediction*. In SIGKDD, 2018.

---

### Meta-Review · Area_Chair_Z8No · 2026-01-08

**Summary:**

This paper addresses the important problem of embedding rank collapse in deep recommender systems and proposes RoBlock, a modular architecture that integrates rank-1 normalization, HSIC-based decoupling, field-wise routing, and residual connections. Reviewers appreciated the clear motivation, coherent system design, and the alignment between theoretical analysis and empirical results showing improved rank preservation under depth and width scaling. However, concerns remain about limited novelty, as most components are adaptations of existing ideas, and the contribution largely lies in their integration. In addition, the paper lacks sufficient discussion and evaluation of computational efficiency, practical deployment cost, and comparisons to simpler collapse-mitigation baselines. The empirical validation is also limited to a narrow set of datasets, raising questions about generality.

**Reviewer Concerns:**

Reviewers have concerns about incremental novelty, lack of comparisons to more baselines, and evaluation on a limited set of data.

**Reviewer Scores:**

One reviewer mentions changing the score to borderline (but not reflected in the final score). Others keep their scores.

---

### Decision · Program_Chairs · 2026-01-26

Reject